# Learning Topology-Aware Representations via Test-Time Adaptation for Anomaly Segmentation

## Abstract

Test-time adaptation (TTA) has emerged as a powerful paradigm for handling distribution shifts in deep models, particularly for anomaly segmentation, where pixel-wise labels of anomalous regions are typically unavailable during training. We introduce TopoTTA (Topological Test-Time Adaptation), a novel framework that incorporates persistent homology, a tool from topological data analysis, into the TTA pipeline to enforce structural consistency in segmentation. By applying multi-level cubical complex filtration to anomaly score maps, TopoTTA generates robust topological pseudo-labels that guide a lightweight test-time classifier, enhancing binary segmentation quality without retraining the backbone model. Our method eliminates the need for heuristic thresholding and generalises across both 2D and 3D modalities. Extensive experiments on MVTec AD and BraTS datasets demonstrate significant improvements over state-of-the-art unsupervised anomaly detection and segmentation methods in terms of F1 score, particularly on anomalies with complex geometries.

## 1 Introduction

Test-time adaptation has become a crucial strategy in enabling deep models to generalise beyond training distributions, especially in scenarios where labelled data is scarce or unavailable at deployment [1, 2]. A particularly relevant application is anomaly segmentation (AS), where the goal is to identify fine-grained, pixel-level anomalies in test images, typically without access to annotated anomalous examples during training [3]. In such settings, anomaly detection and segmentation (AD&S) models produce spatial anomaly score maps that must be binarised into segmentation masks [4]. However, this binarisation often depends on thresholds learned from nominal (normal) data, leading to poor generalisation across object types and anomaly patterns [5, 6, 7].

While supervised AD methods [8, 9, 10, 11] have shown strong performance, they demand large-scale annotated datasets, which are impractical for rare or heterogeneous anomalies [12, 13]. This challenge has motivated unsupervised approaches that rely on nominal data alone. Yet, these models often rely on static score thresholding and struggle with structure-preserving segmentation under test-time distribution shifts.

Test-Time Training (TTT) has recently emerged as a promising unsupervised learning technique that allows models to adapt to test samples using auxiliary self-supervised tasks during inference [1, 14, 15, 16, 17]. Initially developed in the context of domain adaptation and generalisation, TTT dynamically adjusts a model's representations to the test data without requiring access to the source distribution or labels [18]. This work explores how TTT can be enhanced by incorporating strong inductive biases that encourage structural consistency and training a test-time contrastive encoder that adapts feature representations for improved binary segmentation.

Submitted to 39th Conference on Neural Information Processing Systems (NeurIPS 2025). Do not distribute.

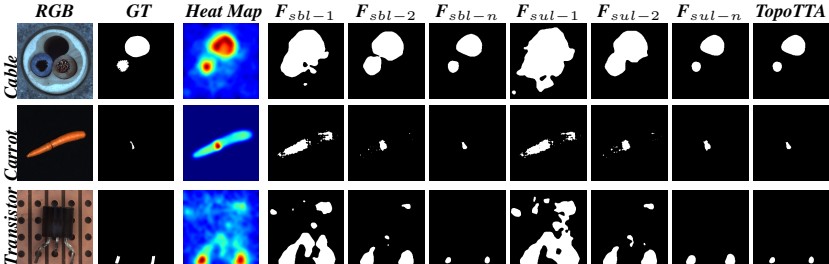

Figure 1: Progressive refinement of anomaly segmentation using multi-level filtrations on cubical complexes. Each row shows a 2D or 3D test image with (left to right): RGB input, ground truth (GT), anomaly heatmap, binary masks from sublevel and superlevel filtrations ($F_{\text{sbl-*}}$, $F_{\text{sul-*}}$), and the final TopoTTA output. Filtrations extract persistent topological features to guide robust segmentation refinement via the PCES module.

Topological Data Analysis (TDA) provides a complementary perspective for understanding high-dimensional data by extracting persistent structural features [19]. Tools such as persistent homology (PH) [20] can quantify the connectedness and saliency of components in an anomaly score map, without relying on prior assumptions about the anomaly's appearance or shape. We propose leveraging TDA as an inductive prior during test-time to refine segmentation by encoding multi-scale spatial structures in the anomaly signal.

TopoTTA presents a framework that integrates topological priors into the TTT paradigm to improve pixel-wise AS. Our approach enhances pseudo-label reliability by leveraging cubical complex filtrations applied to anomaly score maps, enabling persistent structural features to guide binary segmentation mask refinement, as depicted in Figure 1. These topologically derived pseudo-labels are then used to supervise a pixel-level contrastive encoder, trained on-the-fly using features extracted from a frozen pre-trained backbone. This contrastive learning strategy encourages the encoder to align features within anomalous and nominal regions while maximally separating them across the refined pseudo-label space. By incorporating multi-level sub and super-level filtration, without requiring handcrafted thresholds, our method introduces a topology-aware adaptation mechanism at inference time that generalises across datasets and domains. To the best of our knowledge, this is among the first works to integrate persistent homology-based multi-scale topological filtering directly into a test-time learning framework for pixel-wise anomaly segmentation. Our contributions are as follows:

- We present a theoretically grounded test-time adaptation method that extracts persistent structural priors via multi-level cubical complex filtrations. These topological features, including connected components and holes, are provably stable under perturbations to the anomaly score map.

- We propose a pixel-level contrastive encoder (PCES), trained at inference using sparse pseudo-labels derived from topological persistence, to produce dense, structurally consistent segmentation without requiring access to source data or retraining.

- Our framework is modular and model-agnostic, functioning as a plug-in refinement module for any AD&S method that outputs an anomaly score map, and generalises across 2D, 3D, and medical imaging domains.

TopoTTA achieves up to **28.8%** F1 improvement over existing test-time and unsupervised segmentation baselines on the MVTec AD, MVTec 3D-AD, and BraTS 2021 datasets. Ablation studies confirm the individual benefits of persistent filtration, topological supervision, and contrastive feature alignment across modalities and architectures.

## 2 Related Work

AS under distribution shift presents a complex challenge that spans multiple research domains. Addressing this requires (1) detecting fine-grained anomalies in the absence of supervision, (2) encoding meaningful structural priors, and (3) adapting models to unseen test-time distributions. In this section, we review relevant work on unsupervised AD&S, TDA in vision, and TTT, and

highlight how our method bridges gaps across these areas. Specifically, our approach integrates PH-based topological priors with per-instance TTT to refine binary segmentation masks using structural information extracted from AS maps.

## 2.1 Unsupervised Anomaly Detection and Segmentation (AD&S)

Unsupervised AD&S has gained traction due to its ability to identify anomalies in the absence of annotated data [21]. Early methods relied on reconstruction-based models such as autoencoders [22, 23, 24, 25, 26], inpainting [27, 28, 29, 30, 31], and diffusion models [32, 33, 34], assuming that nominal patterns can be reliably reconstructed while anomalies cannot. However, these approaches tend to produce blurry reconstructions or overfit to nominal structures, limiting their efficacy under distribution shift. Feature-based methods compare test sample embeddings to nominal training data [35, 36, 37], while teacher-student frameworks [38, 39, 40, 41, 42] introduce inductive bias through cross-network consistency. These approaches enhance robustness but depend on global similarity metrics, often missing local structural discrepancies, limiting their effectiveness in dense tasks like segmentation. Alternative strategies use generative priors via normalizing flows [43, 44, 45, 46] or synthetic anomalies within one-class classification [47, 48, 49, 50], yet they typically fall short in spatial resolution or adaptability for pixel-level accuracy.

Recent techniques such as PatchCore [36] and PaDiM [37] leverage pre-trained vision transformers and memory banks for strong feature representation, but rely on fixed distance metrics and heuristic thresholds, making them sensitive to class-specific tuning and lacking in structural sensitivity.

*In contrast, our proposed method leverages topological information extracted directly from anomaly score maps to guide refinement.* By applying cubical complex filtrations, we identify persistent structural features such as mathematical holes and connected components, which serve as robust pseudo-labels for pixel-level adaptation. This goes beyond purely statistical or embedding-based comparisons by embedding inductive structural priors into the segmentation process.

## 2.2 Topological Data Analysis in Image Segmentation

TDA, particularly through PH, has been increasingly used in medical image analysis to capture shape descriptors and multi-scale structural features [51, 52, 53, 54, 55]. Applications include tumour classification, liver lesion detection, and neuronal morphology analysis. However, most of these works are confined to offline analysis or post-hoc characterisation. They are typically not integrated with modern learning frameworks nor applied in settings with severe domain shift or test-time adaptation.

To our knowledge, *no prior work integrates PH via multi-level cubical filtrations into a test-time learning pipeline for pixel-level anomaly segmentation*. Unlike prior applications of TDA, which operate on static representations, our method uses TDA as a dynamic, learnable signal for refining segmentation masks at inference. This enables principled pseudo-label generation that reflects persistent topological features and improves robustness under noise and uncertainty.

## 2.3 Test-Time Training (TTT)

TTT has emerged as an effective strategy for adapting pre-trained models to unseen distributions using only test-time data [56]. TTT has shown promise in classification [57], semantic segmentation [58], and object detection [59], particularly under domain shift. TTT approaches vary in adaptation granularity, ranging from batch-level [60], online [61], to per-instance [62] settings, and typically rely on self-supervised losses or consistency constraints during inference.

Recently, TTT4AS [63] extended this paradigm to AS. It proposes training a per-image support vector machine (SVM) classifier at test time, using pseudo-labels generated from high-scoring anomaly regions via non-maximum suppression and neighbourhood enrichment. This enables flexible adaptation without backpropagation, using sparse but discriminative features to improve binary mask prediction. However, TTT4AS depends on heuristics for peak selection and local smoothing, lacks explicit structural reasoning, and can produce inconsistent masks under noise or geometric anomalies.

In contrast, our method, *TopoTTA*, introduces a topologically-informed TTA mechanism. We replace heuristic-based pseudo-labelling with PH computed via multi-scale cubical filtrations of the

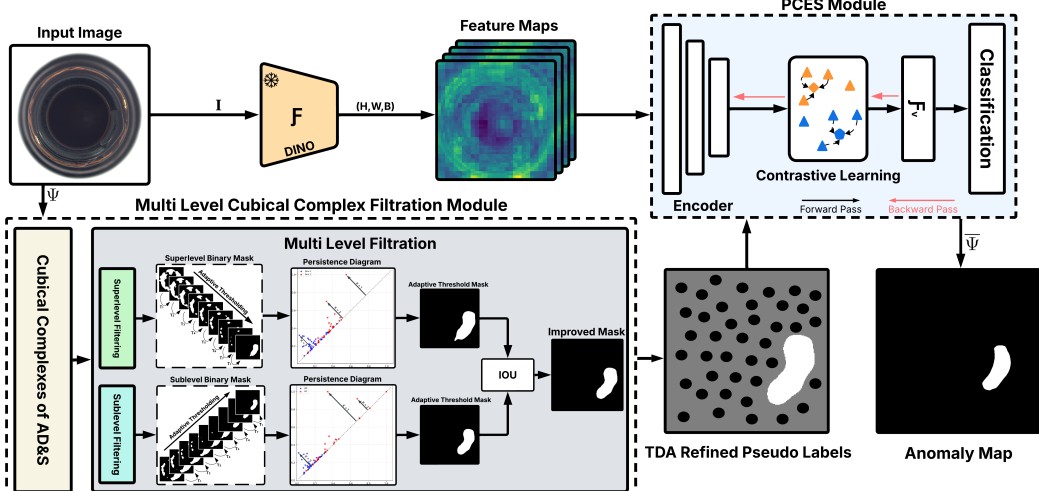

Figure 2: Overview of the **TopoTTA** architecture. Given a test image $I$, an AD&S method produces an anomaly score map $\Psi$. A pre-trained feature extractor $F$ generates dense feature maps from $I$. Topological pseudo-labels are extracted by applying multi-level cubical complex filtrations (both sublevel and superlevel) to $\Psi$, producing structurally meaningful binary masks via persistent homology. These masks are fused using IoU to generate sparse pseudo-labels. A lightweight classifier is then trained on selected feature points from $F(I)$ using these labels and applied across the full feature map to produce a refined binary anomaly segmentation. This test-time adaptation pipeline exploits both intensity-based cues and topological structure to improve segmentation robustness and generalisation.

AS map. This allows us to extract geometrically and topologically consistent pseudo-labels that reflect connectedness, holes, and persistent structures. These are used to supervise a lightweight contrastive classifier trained at test time. To the best of our knowledge, TopoTTA is the first method to incorporate persistent topological priors into the TTT pipeline for anomaly segmentation, improving generalisation and structural precision across both 2D and 3D modalities.

## 3 TTA Using Multi-Level Topological Filtering

Building on the limitations of prior test-time training approaches, particularly their reliance on heuristic pseudo-labels and lack of structural awareness, we propose **TopoTTA**, a topology-guided adaptation framework for AS. Our method replaces intensity-based thresholds and local peak heuristics with persistent topological descriptors extracted via multi-level cubical filtrations of the anomaly score map. Our approach operates as a model-agnostic, downstream enhancement module that can be integrated with any AD&S method, producing a per-pixel anomaly score map. Given a test sample $I$, its anomaly score map $\Psi$, and dense feature representations $F$ extracted via a frozen general-purpose backbone, TopoTTA constructs sparse pseudo-labels using multi-level topological filtration of $\Psi$. These pseudo-labels supervise a lightweight classifier trained on a subset of spatial features from $F$, which is then applied across the full feature map to predict a refined binary anomaly segmentation mask. This design allows TopoTTA to exploit both anomaly-localised signal and global topological structure at test time, without requiring retraining or backpropagation through the backbone network. The full adaptation pipeline, including topological filtration, classifier training, and final mask refinement, is illustrated in Figure 2.

### 3.1 Multi-Level Cubical Complex Filtration

The *Multi-Level Cubical Complex Filtration Module*, shown in Figure 2, is a central module in TopoTTA that extracts stable topological priors from anomaly score maps. This block proceeds in two logical stages. First, a cubical complex is constructed from the anomaly score map $\Psi$; then, multi-level filtration is applied to generate persistence diagrams that inform robust pseudo-labels.

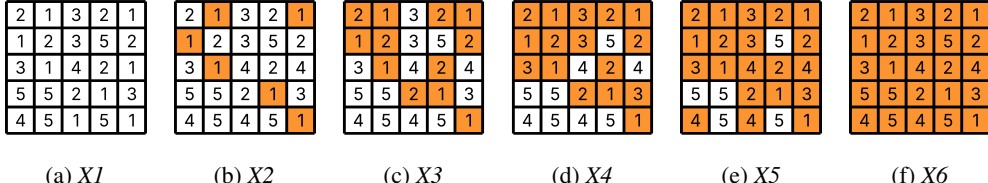

(a) *X1*     (b) *X2*     (c) *X3*     (d) *X4*     (e) *X5*     (f) *X6*

Figure 3: Sublevel and superlevel filtrations on a 2D grayscale image. Given a grayscale image $X$ with pixel intensities $I_{ij} \in [0, 255]$, a **sublevel filtration** constructs nested binary masks $X_1 \subseteq \cdots \subseteq X_T$ by including pixels satisfying $I_{ij} \leq \tau_t$ at increasing thresholds $\tau_1 < \cdots < \tau_T$. Conversely, a **superlevel filtration** includes pixels with $I_{ij} \geq \tau_t$ using decreasing thresholds. These filtrations capture evolving topological features such as connected components and holes.

### 3.1.1    Cubical Complex Construction

To encode spatial structure, we treat the 2D anomaly score map $\Psi \in \mathbb{R}^{H \times W}$ as a discrete topological space, forming a *cubical complex* $\mathcal{K}$. We adopt cubical complexes due to their natural alignment with grid-structured image data, enabling efficient computation without triangulation overhead, as supported by Bleile *et al.* [64] and Rieck *et al.* [65]. Each pixel defines a 0-cell (point), and neighbouring pixels define higher-dimensional elements: 1-cells (edges), 2-cells (squares), and, in 3D, 3-cells (voxels). This structure captures the adjacency and continuity of intensity patterns in $\Psi$. By ensuring that all lower-dimensional faces of each cube are included, the complex is closed under subcells and ready for topological analysis. A detailed mathematical formulation is mentioned in Appendix A.1.

### 3.1.2    Multi-Level Topological Filtration

To extract shape features from $\Psi$, we define a filtration function $f : \mathcal{K} \to \mathbb{R}$ that assigns each cube a scalar value based on the maximum intensity of its vertices. To perform multi-level filtration, we construct two complementary filtrations over the cubical complex $\mathcal{K}$ derived from the anomaly score map $\Psi$. In the *sublevel filtration* [65], denoted as $K(a_i) = \{\sigma \in \mathcal{K} \mid f(\sigma) \leq a_i\}$, cubes are added progressively based on increasing threshold values, thereby accumulating low-intensity regions first. Conversely, the *superlevel filtration* [64], defined as $K^\uparrow(b_i) = \{\sigma \in \mathcal{K} \mid f(\sigma) \geq b_i\}$, begins with high-intensity areas and progressively includes regions of decreasing intensity. This bidirectional filtration process allows TopoTTA to capture anomaly structures that may manifest across both ends of the intensity spectrum. As illustrated in Figure 3, these filtrations form a nested sequence of subcomplexes that reflect the evolving topological structure of $\Psi$ under varying thresholds, enabling robust identification of persistent features such as connected regions and hollow defects.

PH [66] is then computed across these filtrations, yielding birth-death pairs $(b_\sigma, d_\sigma)$ for topological features (connected components, holes). The **persistence** $d_\sigma - b_\sigma$ quantifies the feature's significance. These are summarised in persistence diagrams.

To form pseudo-labels, we retain the most persistent features (Top-K or those exceeding a threshold $\tau$), and create binary masks from both filtration types. The masks are fused using intersection-over-union (IoU) to eliminate spurious or short-lived artefacts:

$$\mathrm{IoU}(K(a_i), K^\uparrow(b_i)) = \frac{|K(a_i) \cap K^\uparrow(b_i)|}{|K(a_i) \cup K^\uparrow(b_i)|}.$$

These structurally meaningful masks form the sparse supervision used in downstream test-time classifier training. This filtration-based approach enables us to identify topologically salient features that are resilient to noise and threshold perturbations, an aspect we formalise in the following subsection through a stability analysis grounded in persistent homology theory.

## 3.2 Theoretical Justification: Stability of Topological Pseudo-Labels

Denote by $I : \Omega \to \mathbb{R}$ an anomaly score map over a discrete image domain $\Omega \subset \mathbb{Z}^2$, and let $K$ be the corresponding cubical complex, where each image pixel defines a 0-cell, and adjacent pixels define higher-dimensional cells (edges, squares).

We define a filtration function $f : K \to \mathbb{R}$ over the complex by assigning:

$$f(\sigma) = \max_{p \in \sigma} I(p),$$

where $\sigma$ is any cell in the cubical complex and $p \in \sigma$ denotes a pixel vertex of the cell.

This induces a sublevel filtration:

$$K_\alpha := \{\sigma \in K \mid f(\sigma) \leq \alpha\},$$

which we use to compute persistent homology $\mathrm{PH}_k(K, f)$ in dimension $k$, yielding a persistence diagram or barcode $\mathcal{B}_k = \{(b_i, d_i)\}_i$.

**Lemma 1** (Topological Stability of Anomaly Structures). *Let $f, g : K \to \mathbb{R}$ be two filtration functions derived from anomaly score maps $I$ and $\tilde{I}$, such that $\|f - g\|_\infty \leq \varepsilon$. Then, for every homology dimension $k$, the bottleneck distance between the corresponding persistence diagrams is bounded by:*

$$d_B\left(\mathrm{PH}_k(K, f), \mathrm{PH}_k(K, g)\right) \leq \varepsilon.$$

*Proof.* This result follows directly from the classical stability theorem in persistent homology [67]. Given two tame filtration functions $f$ and $g$ over the same complex $K$, their persistence diagrams satisfy:

$$d_B\left(\mathrm{PH}_k(K, f), \mathrm{PH}_k(K, g)\right) \leq \|f - g\|_\infty,$$

where $d_B$ denotes the bottleneck distance and $\|\cdot\|_\infty$ is the supremum norm over the domain of filtration functions. $\square$

**Implication:** This lemma guarantees that small variations in anomaly score maps, due to noise or uncertain model outputs, result in only small changes to the extracted topological features. Hence, persistent structures with long lifespans (large $d_i - b_i$) are robust to such perturbations, making them reliable candidates for pseudo-labels in test-time adaptation. This provides a principled justification for our use of persistent homology to refine segmentation masks during inference.

## 3.3 Pixel-Level Contrastive Encoder for Binary Segmentation (PCES)

To achieve precise pixel-level anomaly localisation, particularly within a TTA framework, we introduce a lightweight contrastive Multi-Layer Perceptron (MLP) encoder, denoted $E_\theta(\cdot)$. The fundamental purpose of this module is to leverage the spatially-resolved pseudo-anomaly scores $\Psi \in \mathbb{R}^{H \times W}$, derived from our TDA pipeline, as a dynamic supervisory signal. This supervision guides the training of $E_\theta(\cdot)$ to refine the initial dense feature maps $F \in \mathbb{R}^{H \times W \times B}$ into a more discriminative representation tailored for segmentation. We choose a shallow MLP architecture to ensure fast per-image optimisation and to avoid overfitting on sparse pseudo-labels at test time. This design balances segmentation accuracy and computational efficiency, aligning with recent findings in single-image test-time adaptation [62, 63].

The encoder $E_\theta(\cdot)$ possesses a shallow architecture, comprising three sequential linear transformation layers, each followed by a Gaussian Error Linear Unit (GeLU) activation function. For every spatial location $i$ in the input image, $E_\theta(\cdot)$ takes the corresponding feature vector $f_i \in \mathbb{R}^B$ from $F$ and projects it into a latent embedding space, yielding an embedding $z_i = E_\theta(f_i)$. The core design principle is to structure this embedding space such that feature vectors originating from image regions that exhibit similar topological characteristics (as indicated by the TDA-derived scores in $\Psi$) are mapped to proximate locations in the latent space.

This targeted embedding space organisation is realised by optimising the parameters $\theta$ of $E_\theta(\cdot)$ through a formulated contrastive loss function. Given a pair of embeddings $(z_i, z_j)$, where $z_k = E_\theta(f_k)$, the loss is defined as:

$$\mathcal{L}_{\text{contrastive}} = \mathbb{E}_{(z_i, z_j, y_{ij})} \left[ (1 - y_{ij}) \cdot d(z_i, z_j)^2 + y_{ij} \cdot \max(0, m - d(z_i, z_j))^2 \right] \tag{1}$$

In the above formulation, $d(z_i, z_j)$ denotes the Euclidean distance $\|z_i - z_j\|_2$, and $m > 0$ is a pre-defined margin ($m = 1.0$) that dictates the desired separation for dissimilar pairs. The binary label $y_{ij} \in \{0, 1\}$ governs the loss's behaviour and is derived from the TDA-refined pseudo-labels. Specifically, $y_{ij} = 0$ is assigned to "similar pairs," where the input features $f_i$ and $f_j$ correspond to regions consistently identified by $\Psi$ (both are strongly indicated as nominal, or both as anomalous, based on appropriate thresholding of $\Psi_i$ and $\Psi_j$). For such pairs, the loss simplifies to $d(z_i, z_j)^2$, encouraging their embeddings $z_i$ and $z_j$ to converge. Conversely, $y_{ij} = 1$ is assigned to "dissimilar pairs," where $f_i$ and $f_j$ originate from regions with contrasting TDA-derived characteristics. For these pairs, the loss becomes a repulsive term $\max(0, m - d(z_i, z_j))^2$, penalising instances where their embeddings are closer than the margin $m$ and thus promoting their separation.

This contrastive training process is executed at test-time for each input image, allowing $E_\theta(\cdot)$ to adapt to the specific content of that image. Upon convergence of this TTA optimisation, the encoder $E_\theta(\cdot)$ effectively transforms the original feature map into the structured, discriminative embedding space. The final dense binary segmentation mask $\Psi' \in \{0, 1\}^{H \times W}$ is then generated by applying a simple distance-based classifier to these learned embeddings $z_k$.

# 4 Results & Discussions

We evaluate *TopoTTA* on three benchmark datasets spanning both 2D and 3D modalities. In the 2D domain, we used MVTecAD [68], an industrial anomaly detection dataset, and BraTS 2021 [69], which provides brain tumor segmentation data. For 3D evaluation, we used the 3D MVTecAD dataset [70]. We utilise PatchCORE [71] as the backbone for feature extraction and anomaly scoring in 2D, while for 3D, we adopt M3DM [72] and CMM [73] models. For binary segmentation, the general practice among baseline papers [71, 72, 73] is to determine thresholding parameters based on the statistical characteristics of the validation set. In this work, we followed the same approach as proposed in [63], where a thresholding ($\mu + c\sigma$) is applied to generate binary segmentation. For model evaluation, we employed classification metrics including Precision, Recall, and the F1 Score to assess the performance of the baseline models quantitatively. All experiments have been executed on a single NVIDIA GeForce RTX 3080 Ti GPU. The comprehensive experimental setting is mentioned in Appendix A.2. Anomaly classes are listed vertically, and binary segmentation maps across various baselines are reported horizontally in Tables 1, 2 & 3. A time complexity analysis is provided in Appendix A.6, with additional discussion on limitations and future directions in Appendix A.7.

## 4.1 2D AS&D

We evaluated our proposed architecture on the 2D MvTec AD dataset [68], as shown in the qualitative binary maps in Figure 4 and quantitative results in Table 1. The baseline method [71] is trained using DINOv2-extracted features, and its optimal threshold is determined based on $\mu + 3\sigma$ and TTT4AS [63] through local maxima detection. Our approach significantly outperforms the baseline methods in terms of mean Precision, Recall, and F1 Score. Specifically, compared to THR [71], we observed improvements of **16.8%** in Precision, **18.3%** in Recall, and **28.0%** in F1 Score. Additionally, when comparing TopoTTA with TTT4AS [63], we achieved improvements of **15.0%** in mean Precision, **4.2%** in mean Recall, and **9.7%** in mean F1 Score. These results demonstrate that our method surpasses existing approaches both quantitatively and qualitatively, achieving superior performance across all evaluation metrics. The generalisation performance of TopoTTA has been further validated on the BraTS 2021 brain tumour segmentation dataset [69]. Quantitative results indicate that our proposed method (Precision: 0.468, Recall: 0.586, F1: 0.457) outperforms established baselines. Specifically, it surpasses TTT4AS [63] (P: 0.554, R: 0.382, F1: 0.409) by 4.82% in mean F1 Score, and [71] (P: 0.311, R: 0.882, F1: 0.426, using an optimal $\mu + 1\sigma$ threshold for this task) by 3.2% in mean F1 Score. More results, including PaDiM [37] model are shown in Appendix A.4.

## 4.2 3D Multi-Modal AS&D

Our proposed methodology is also being evaluated on the MVTec 3D-AD [70] benchmark dataset, with qualitative segmentation maps depicted in Figure 5 and quantitative performance metrics tabulated in Table 2 & 3. Our TopoTTA approach exhibits significantly enhanced performance relative to these baseline [73, 63] configurations w.r.t mean Precision and F1 Score. Specifically, relative to CMM-THR [73], our method achieved an uplift of **30%** in Precision and **19.3%** in

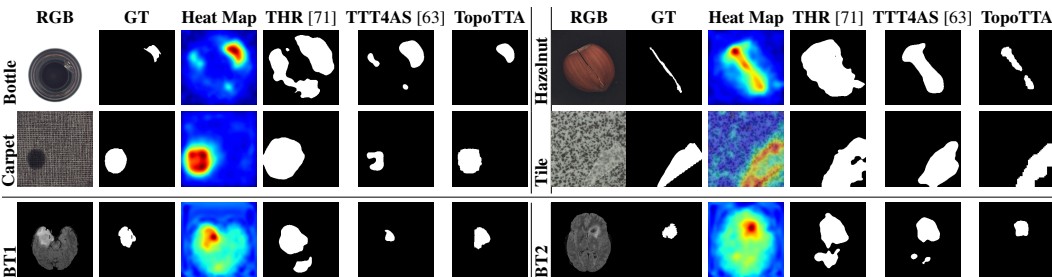

Figure 4: Qualitative comparison of various anomaly detection methods for different objects using PatchCore model on the 2D MvTec AD and BraTs 2021 datasets.

Table 1: Performance evaluation of PatchCore across 15 categories of the MVTec AD dataset and their mean, comparing three binary map strategies: (a) THR ($\mu + 3\sigma$), (b) TTT4AS, and (c) TopoTTA. The table highlights the best result for each Precision, Recall, and F1 Score metric in **bold black** and the second-best in blue.

| Metric | Bottle | Cable | Capsule | Carpet | Grid | Hazelnut | Leather | MetalNut | Pill | Screw | Tile | T-brush | Transistor | Wood | Zipper | Mean |
|---|---|---|---|---|---|---|---|---|---|---|---|---|---|---|---|---|
| **(a) PatchCore - Binary Map - THR ($\mu + 3\sigma$) [71]** |
| Precision | 0.397 | 0.344 | 0.278 | 0.362 | **0.432** | 0.405 | 0.297 | 0.435 | 0.347 | **0.298** | 0.403 | 0.286 | 0.334 | 0.384 | 0.268 | 0.351 |
| Recall | 0.510 | 0.465 | 0.626 | 0.522 | 0.428 | 0.380 | 0.542 | **0.566** | 0.618 | 0.522 | **0.517** | 0.542 | 0.287 | 0.469 | 0.605 | 0.507 |
| F1 Score | 0.175 | 0.194 | 0.085 | 0.092 | 0.078 | 0.120 | 0.045 | 0.311 | 0.188 | 0.066 | 0.209 | 0.123 | 0.114 | 0.121 | 0.119 | 0.136 |
| **(b) PatchCore - Binary Map - TTT4AS [63]** |
| Precision | 0.662 | 0.502 | 0.163 | 0.413 | 0.185 | 0.425 | 0.212 | 0.644 | 0.337 | 0.046 | 0.644 | 0.272 | 0.391 | 0.470 | 0.449 | 0.368 |
| Recall | 0.664 | 0.565 | 0.632 | 0.824 | 0.787 | 0.861 | 0.893 | 0.528 | 0.740 | 0.361 | 0.495 | 0.594 | 0.462 | 0.664 | 0.644 | 0.648 |
| F1 Score | 0.593 | 0.480 | 0.197 | 0.457 | 0.272 | 0.499 | 0.286 | 0.482 | 0.358 | 0.078 | 0.474 | 0.301 | 0.318 | 0.464 | 0.469 | 0.382 |
| **(c) PatchCore - Binary Map - TopoTTA** |
| Precision | 0.731 | 0.587 | 0.352 | 0.580 | 0.361 | 0.431 | 0.328 | 0.750 | 0.354 | 0.261 | 0.749 | 0.376 | 0.591 | 0.541 | 0.784 | 0.519 |
| Recall | 0.685 | 0.738 | 0.813 | 0.743 | 0.762 | 0.826 | 0.938 | 0.522 | 0.646 | 0.887 | 0.453 | 0.661 | 0.452 | 0.634 | 0.585 | 0.690 |
| F1 Score | 0.606 | 0.591 | 0.409 | 0.547 | 0.450 | 0.475 | 0.425 | 0.496 | 0.390 | 0.379 | 0.491 | 0.390 | 0.395 | 0.506 | 0.596 | 0.476 |

F1 Score. Moreover, compared to TTT4AS [63], the proposed TopoTTA yields improvements of **20%** in mean Precision, and **8.8%** in mean F1 Score. Similarly, for the M3DM[72], quantitative metrics are shown in Table 3. Relative to M3DM-THR [72], our TopoTTA method achieved an uplift of **31.11%** in Precision and **28.80%** in F1 Score. Moreover, compared to TTT4AS [63], the proposed TopoTTA yielded improvements of **1.7%** in mean Precision, and **6.5%** in mean F1 Score. These results further substantiate that our TopoTTA method consistently exhibits superior efficacy over these M3DM baseline configurations. More qualitative results are shown in Appendix A.5.

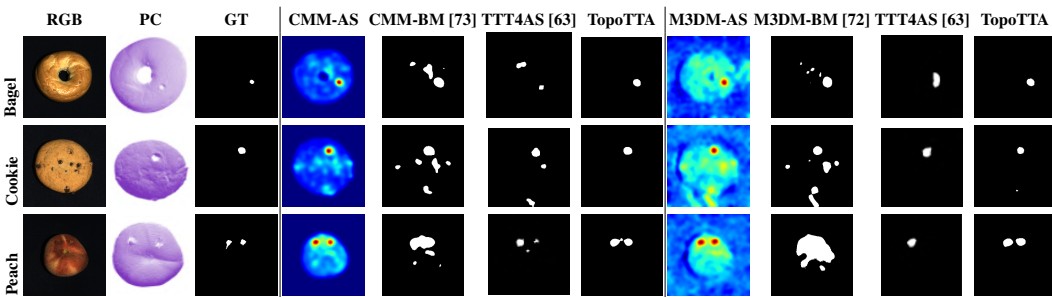

Figure 5: Qualitative comparison of various AD&S methods for different objects using the 3D MvTec AD dataset.

## 4.3 Ablation Study

Table 4 presents the results of an ablation study designed to evaluate the contribution of individual persistence components to anomaly detection performance. Specifically, we analyse the precision, recall, and F1 score when using only the single Top K$^{th}$ farthest persistence component (where K ranges from 1 to 5) derived from features of 2D-PatchCore [71], 3D-CMM [73], and 3D-M3DM [72] models. The results consistently demonstrate the significance of the most persistent topological

Table 2: Performance evaluation of CMM [73] across categories of the MVTec 3D-AD dataset [70].

| Method | Bagel | Gland | Carrot | Cookie | Dowel | Foam | Peach | Potato | Rope | Tire | Mean |
|---|---|---|---|---|---|---|---|---|---|---|---|
| **(b) CMM - THR ($\mu + 3\sigma$) [73]** | | | | | | | | | | | |
| Precision | 0.301 | 0.188 | 0.049 | 0.518 | 0.072 | 0.275 | 0.262 | 0.092 | 0.049 | 0.182 | 0.198 |
| Recall | **0.949** | **0.842** | **0.998** | **0.901** | **0.896** | 0.597 | **0.957** | **0.998** | **0.989** | **0.896** | **0.902** |
| F1 Score | 0.425 | 0.265 | 0.092 | 0.619 | 0.129 | 0.327 | 0.375 | 0.160 | 0.091 | 0.267 | 0.275 |
| **(c) CMM - TTT4AS [63]** | | | | | | | | | | | |
| Precision | 0.432 | 0.258 | 0.242 | 0.713 | 0.195 | 0.214 | 0.353 | 0.252 | 0.264 | 0.111 | 0.303 |
| Recall | 0.745 | 0.766 | 0.889 | 0.603 | 0.739 | **0.732** | 0.872 | 0.888 | 0.865 | **0.904** | 0.800 |
| F1 Score | 0.495 | 0.362 | 0.351 | 0.606 | 0.289 | 0.311 | 0.470 | 0.363 | 0.360 | 0.189 | 0.380 |
| **(d) CMM - TopoTTA** | | | | | | | | | | | |
| Precision | **0.774** | **0.385** | **0.5175** | **0.730** | **0.360** | **0.335** | **0.605** | **0.491** | **0.555** | **0.204** | **0.507** |
| Recall | 0.544 | 0.487 | 0.7519 | 0.685 | 0.358 | 0.559 | 0.702 | 0.712 | 0.415 | 0.625 | 0.740 |
| F1 Score | **0.5726** | **0.391** | **0.567** | **0.646** | **0.324** | **0.384** | **0.547** | **0.527** | **0.437** | **0.290** | **0.468** |

Table 3: Performance evaluation of M3DM [72] across categories of the MVTec 3D-AD dataset [70].

| Method | Bagel | Gland | Carrot | Cookie | Dowel | Foam | Peach | Potato | Rope | Tire | Mean |
|---|---|---|---|---|---|---|---|---|---|---|---|
| **(b) M3DM - THR ($\mu + 3\sigma$)[72]** | | | | | | | | | | | |
| Precision | 0.174 | 0.105 | 0.045 | 0.493 | 0.221 | 0.254 | 0.067 | 0.050 | 0.194 | 0.127 | 0.173 |
| Recall | **0.949** | **0.980** | **0.997** | 0.712 | **0.909** | 0.536 | **1.000** | **0.999** | **0.917** | **0.894** | **0.889** |
| F1 Score | 0.270 | 0.174 | 0.085 | 0.547 | 0.328 | 0.318 | 0.121 | 0.094 | 0.308 | 0.204 | 0.245 |
| **(c) M3DM - TTT4AS[63]** | | | | | | | | | | | |
| Precision | 0.498 | 0.486 | 0.337 | 0.752 | 0.464 | 0.386 | 0.536 | 0.347 | 0.561 | 0.302 | 0.467 |
| Recall | 0.607 | 0.706 | 0.750 | 0.351 | 0.691 | 0.624 | 0.779 | 0.684 | 0.543 | 0.669 | 0.640 |
| F1 Score | 0.478 | **0.525** | 0.422 | 0.443 | **0.514** | **0.440** | **0.585** | 0.419 | 0.468 | 0.383 | 0.468 |
| **(d) M3DM - TopoTTA** | | | | | | | | | | | |
| Precision | **0.595** | 0.369 | 0.425 | 0.844 | 0.522 | 0.395 | 0.580 | 0.354 | 0.592 | 0.404 | 0.484 |
| Recall | 0.579 | 0.748 | 0.809 | 0.540 | 0.767 | **0.831** | 0.803 | 0.862 | 0.842 | 0.874 | 0.732 |
| F1 Score | 0.499 | 0.400 | 0.478 | 0.598 | 0.645 | 0.459 | 0.580 | 0.416 | 0.695 | 0.564 | 0.533 |

feature (**Top1**). Using the **Top1** component yields the highest precision across all three baseline models (0.519, 0.507, and 0.484, respectively). More importantly, the **Top1** component also achieves the highest F1 score for all the three models (0.476, 0.468, and 0.533), indicating the best balance between precision and recall among the individual components tested. For the 3D-CMM model, the **Top1** component uniquely provides the peak performance across all three metrics. Conversely, selecting components progressively closer to the persistence diagram diagonal (increasing $K$ from 1 to 5) reveals a clear trade-off. While recall consistently increases with $K$ (reaching highs of 0.912 and 0.966 for $K = 5$), precision drops sharply. This leads to a monotonic decrease in the F1 score as $K$ increases for all tested models. Single most persistent component (*Top1*) carries the most discriminative information for achieving balanced anomaly detection performance in this setup.

Table 4: Effect of Top-$K$ persistent features on anomaly segmentation.

| Top K$^{th}$ Farthest Persistence Components | | | | | 2D-PatchCore [71] | | | 3D-CMM [73] | | | 3D-M3DM [72] | | |
|---|---|---|---|---|---|---|---|---|---|---|---|---|---|
| Top1 | Top2 | Top3 | Top4 | Top5 | Prec. | Rec. | F1 | Prec. | Rec. | F1 | Prec. | Rec. | F1 |
| ✓ | | | | | **0.519** | 0.690 | **0.476** | **0.507** | **0.740** | **0.468** | **0.484** | 0.732 | **0.533** |
| | ✓ | | | | 0.432 | 0.799 | 0.447 | 0.426 | 0.493 | 0.398 | 0.231 | 0.943 | 0.336 |
| | | ✓ | | | 0.373 | 0.856 | 0.421 | 0.398 | 0.528 | 0.404 | 0.186 | 0.950 | 0.311 |
| | | | ✓ | | 0.333 | 0.891 | 0.393 | 0.429 | 0.556 | 0.400 | 0.152 | 0.960 | 0.234 |
| | | | | ✓ | 0.305 | **0.912** | 0.370 | 0.356 | 0.579 | 0.395 | 0.125 | **0.966** | 0.198 |

Table 5 examines the effect of different components in our Multi-Level Cubical Complex Filtration using the same baselines. We compare sublevel and superlevel filtrations independently, and in combination with IoU fusion and PCES. While superlevel filtration tends to favour recall (e.g., 0.948 and 0.999), it significantly harms precision. In contrast, the full configuration combining both filtrations with IoU and PCES yields the best F1 scores across all models. This outcome validates the efficacy of our integrated multi-level approach and underscores the importance derived from combining these distinct topological perspectives.

Table 5: Comparison of sublevel, superlevel with IoU and PCES modules

| Multi-Level Cubical Complex Filtration | | | | 2D-PatchCore [71] | | | 3D-CMM [73] | | | 3D-M3DM [72] | | |
|---|---|---|---|---|---|---|---|---|---|---|---|---|
| Sublevel | Superlevel | IoU | PCES | Prec. | Rec. | F1 | Prec. | Rec. | F1 | Prec. | Rec. | F1 |
| ✓ | | | ✓ | 0.393 | 0.524 | 0.370 | **0.533** | 0.490 | 0.417 | 0.290 | 0.934 | 0.394 |
| | ✓ | | ✓ | 0.217 | 0.625 | 0.226 | 0.082 | **0.948** | 0.114 | 0.107 | **0.999** | 0.105 |
| ✓ | ✓ | ✓ | ✓ | **0.519** | **0.690** | **0.476** | 0.507 | 0.740 | **0.468** | **0.484** | 0.732 | **0.533** |

# 5 Conclusion

We proposed *TopoTTA*, a TTA framework that incorporates persistent topological features to refine AS in both 2D and 3D modalities. By using multi-level cubical filtrations, our method produces stable, structure-aware pseudo-labels that enhance segmentation accuracy without retraining. Future work will explore integrating topological priors into end-to-end learning, extending the framework to multi-class and temporal settings, and developing uncertainty-aware filtration strategies to improve robustness under extreme distribution shift.

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

# A Appendix

- 

## A.1 Cubical Persistence

A *primitive interval* $J \subset \mathbb{R}$ is defined as $J = [k, k+1]$ for some $k \in \mathbb{Z}$, referred to as a unit cell (1-cube). This degenerate case $[k]$, where $k \in \mathbb{Z}$, represents a *point cell (0-cube)*. The standard unit interval $J = [0, 1]$ serves as a *unit interval*. A *d-dimensional elementary cube* $C$ is constructed by taking the Cartesian product of a finite set of *basic intervals*:

$$C = J_1 \times J_2 \times \cdots \times J_d \in \mathbb{R}^d, \tag{2}$$

The elementary cubes in a 3D grid consist of vertices, edges, squares (2-cubes), and voxels (3-cubes). The *boundary* of a basic interval $J = [k, k+1]$ consists of its two endpoints: $\partial J = \partial[k, k+1] = [k+1, k+1] - [k, k] = \{k, k+1\}$ which defines the 0-dimensional boundary points (vertices) of the interval. For a d-dimensional elementary cube $C = J_1 \times \cdots \times J_d$, its boundary is made up of all $(d-1)$-dimensional faces and is computed as:

$$\partial C = \sum_{i=1}^{d} (-1)^{i+1} \cdot (J_1 \times \cdots \times \partial J_i \times \cdots \times J_d), \tag{3}$$

where applying $\partial J_i$ replaces the $i^{\text{th}}$ interval with its vertex representation. This ensures that the boundary of $C$ includes all lower-dimensional cubes that form its geometric skeleton. For two elementary cubes $C$ and $C'$, we define $C$ to be a subcube of $C'$, denoted $C \sqsubseteq C'$, if each interval defining $C$ is contained within the corresponding interval of $C'$, that is, $J_i \subseteq J'_i$ for all $i = 1, \ldots, d$. In this case, $C'$ is referred to as a supercube of $C$. Similarly, any cube $P$ that contains $C$ as a subcube is called a *coface* of $C$.

A *cubical complex* $\mathcal{K}$ is a collection of elementary cubes that satisfies two fundamental conditions. First, if a cube $C$ belongs to $\mathcal{K}$, then all its subcubes (lower-dimensional faces) must also be included in the complex; that is, for any cube $P \sqsubseteq C$, it follows that $P \in \mathcal{K}$. Second, if $C \in \mathcal{K}$, all of its boundary components—its $(d-1)$-dimensional faces—are also elements of $\mathcal{K}$. These properties ensure that the complex maintains structural coherence across dimensions. Intuitively, a cubical complex represents a discretized grid as a hierarchical structure composed of geometric entities at multiple levels: 0-cubes (points), 1-cubes (edges), 2-cubes (squares), and 3-cubes (volumetric units), each corresponding to different dimensional cubes is shown in Figure 6.

A map $g : K \to L$ is called a *cubical map* if it preserves the subcube relation. That is, for any two cubes $C, C' \in K$, whenever $C \subseteq C'$, it holds that $g(C) \subseteq g(C')$ in $L$.

An *n-chain* is defined as a formal finite linear combination of $n$-dimensional cubes with integer coefficients. The *chain group* $C_n(K)$ is the free Abelian group generated by all $n$-dimensional cubes in the cubical complex $K$. For a given complex $K$, the associated *cubical chain complex* $C_*(K)$ is represented as a sequence of chain groups connected by boundary operators: These boundary operators satisfy the fundamental property: $\partial_n \circ \partial_{n+1} = 0$

$$\cdots \xrightarrow{\partial_{n+1}} C_n(K) \xrightarrow{\partial_n} C_{n-1}(K) \xrightarrow{\partial_{n-1}} \cdots \xrightarrow{\partial_1} C_0(K) \to 0$$

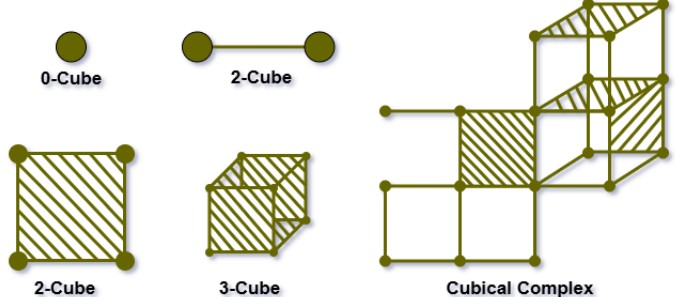

Figure 6: Elementary complexes of different dimension and an exemplary cubical complex.

For a cubical chain complex $C_*(K)$, an $n$-chain $z \in C_n(K)$ is referred to as a *cycle* if it satisfies $\partial_n(z) = 0$, meaning it has no boundary. Since every boundary is itself a cycle by definition, the group of boundaries $B_n(K)$ is a subgroup of the cycle group $Z_n(K)$. These groups are formally defined as:

$$Z_n(K) := \ker(\partial_n) = \{c \in C_n(K) \mid \partial_n(c) = 0\}, B_n(K) := \text{im}(\partial_{n+1}) = \{\partial_{n+1}(c) \mid c \in C_{n+1}(K) \quad (4)$$

The quotient group $H_n(K) = Z_n(K)/B_n(K)$ defines the *$n$-th homology group*, which captures topological features such as $n$-dimensional voids or holes in the complex. The collection gives the full homology of the cubical complex $K$:$H_*(K) = \{H_n(K)\}_{n\in\mathbb{Z}}$.

A *filtration function $f_K : K \to \mathbb{R}$* governs the progressive construction of a cubical complex by assigning to each $d$-cube the first threshold at which it becomes active. This ensures that any cube appears no earlier than its faces: for all cubes $P \sqsubseteq Q$, it holds that $f_K(P) \leq f_K(Q)$. Given this function, we define both *sublevel* and *superlevel* sets corresponding to thresholds $a_i \in \mathbb{R}$. The *sublevel set $K(a_i)$* is defined as:

$$K(a_i) := f_K^{-1}((-\infty, a_i]), \quad \emptyset = K(a_0) \subseteq K(a_1) \subseteq \cdots \subseteq K(a_n) \quad (5)$$

which contains all cubes whose filtration values are less than or equal to $a_i$, forming a nested sequence under increasing thresholds: Similarly, the *superlevel set $K^\uparrow(b_i)$* captures cubes with filtration values greater than or equal to a descending sequence of thresholds $b_i \in \mathbb{R}$, and is defined as:

$$K^\uparrow(b_i) := f_K^{-1}([b_i, +\infty)), \quad \emptyset = K^\uparrow(b_0) \supseteq K^\uparrow(b_1) \supseteq \cdots \supseteq K^\uparrow(b_n) \quad (6)$$

where higher intensity cubes are activated first.

Any cubical inclusion from $K_i$ to $K_j$, where $i \leq j$, induces a linear map between their corresponding homology spaces. This map, denoted as: $\varphi_{ij} : H_k(K_i) \to H_k(K_j)$, captures how topological features evolve across the filtration due to the *functoriality* property of homology. When applying Eq. 5, we obtain an ordered sequence of homology groups connected by these induced maps:

$$H_k(K_0) \xrightarrow{\varphi_{01}} H_k(K_1) \xrightarrow{\varphi_{12}} \cdots \xrightarrow{\varphi_{n-1,n}} H_k(K_n) \quad (7)$$

The Eq.7 forms a persistence module: $\mathcal{P} = \{H_k(\mathcal{K}_i), \phi_{ij}\}_{0 \leq i \leq j \leq n}$, which defines the $k^{\text{th}}$ cubical persistent homology. It tracks how $k$-dimensional topological features (e.g., holes) appear and disappear across the filtration, assigning to each feature $\sigma$ a birth time $b_\sigma$ and death time $d_\sigma$. The lifespan $d_\sigma - b_\sigma$ quantifies the persistence of $\sigma$, and the set of all such intervals $[b_\sigma, d_\sigma)$ constitutes the *persistence barcode*. The $k^{\text{th}}$ *persistence diagram* ($\text{PD}_k(\mathcal{K})$) consists of all birth-death pairs $(b_\sigma, d_\sigma)$ such that $\sigma \in H_k(\mathcal{K}_i)$ for $b_\sigma \leq i < d_\sigma$. These diagrams are represented as multisets of points in $\mathbb{R}^2$, where each point encodes the birth and death times of a topological feature. Due to their irregular structure, PDs are not directly compatible with standard machine learning pipelines [74]. Hence, they are often mapped to fixed-dimensional representations through *vectorisation* — a process defined as a function $\Phi : \text{PD} \to \mathbb{R}^M$, enabling seamless integration with ML models.

## A.2 Experimental Settings

We applied a CNN-based WideResNet-50 [75] and a transformer-based DINO [76] backbone as feature extractors ($F$) for the 2D MVTec AD [68], and BraTS 2021 [69] dataset, while the 3D

MVTec dataset leveraged DINO-v2[77] and Point-MAE [78] backbones. Our proposed architecture is benchmarked against state-of-the-art AD&S methods, including memory bank-driven approaches like PatchCore [71] (RGB) and M3DM [72] (RGB + 3D point clouds), alongside reconstruction-based techniques such as CMM [73]. The 2D RGB-based MVTec AD [68] dataset encompasses 15 object categories, comprising 5,354 training images and 1,725 test images. Each class contains normal (defect-free) samples for training, while test sets include both normal and anomalous instances with diverse defect types, accompanied by ground truth annotations. The BraTS 2021 segmentation dataset is used for binary classification (normal vs. tumor) in brain tumor analysis. It contains recorded data from 2040 patients [69]. The MVTec 3D-AD [70] data set spans 10 categories, featuring 2,656 nominal training images and 1,197 test samples. Both 2D and 3D datasets are standardised to $224 \times 224$ pixels during preprocessing to ensure input consistency. For test-time training, a lightweight MLP-based encoder using a contrastive learning classifier with a margin of 1.0 is deployed.

## A.3 Binary Maps at Multi-level Cubical Complex Constructions

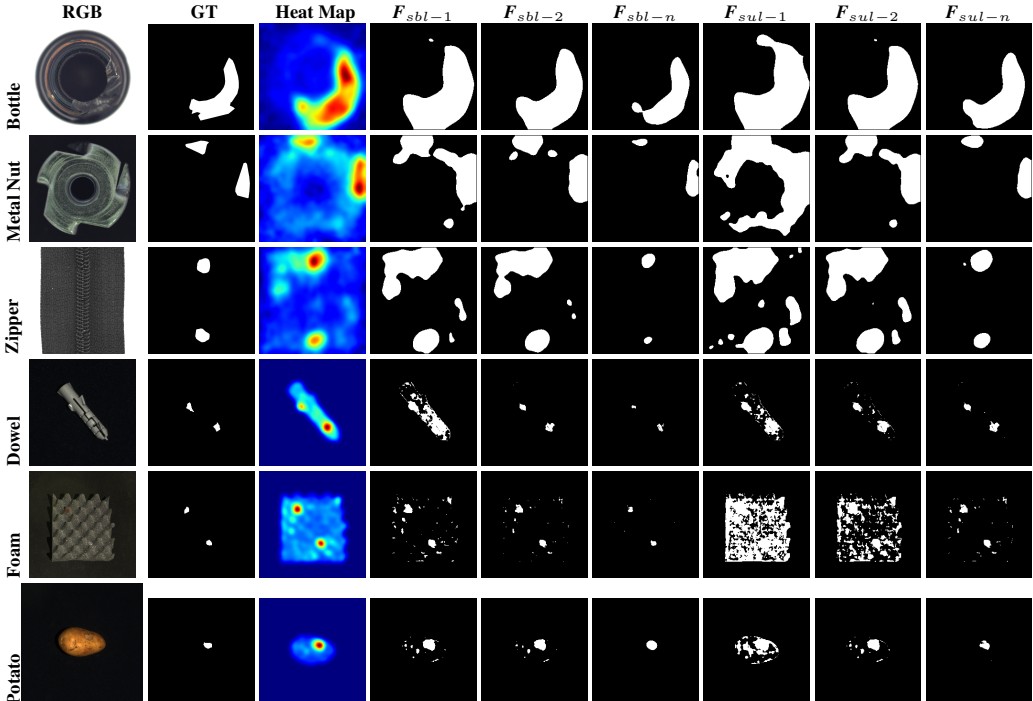

Figure 7: Visualization of the progressive refinement of anomaly localisation across both 2D and 3D modalities. The visualization includes the input RGB image, ground truth, initial heatmap, and the evolution of binary anomaly masks through sub-level ($F_{sbl-1}$ to $F_{sbl-n}$) and super-level ($F_{sul-1}$ to $F_{sul-n}$) filtrations. This stepwise transformation demonstrates how the model effectively suppresses noisy initial detections by applying adaptive thresholding at multiple filtration levels. The resulting refined representations contribute to the improved performance of the *TopoTTA* segmentation, guided by the PCES, highlighting the topological consistency and spatial coherence in anomaly delineation.

 ## A.4 2D AD&S

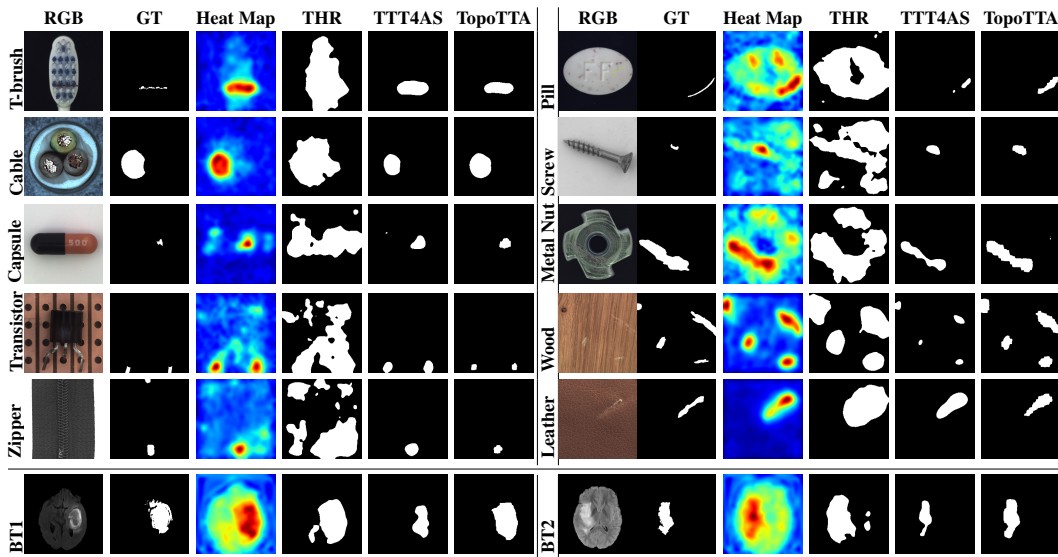

Figure 8: Qualitative comparison of various anomaly detection methods for different objects using PatchCore model on 2D MvTec AD and BraTs 2021 datasets.

Table 6: Performance evaluation of PaDiM [37] across 15 categories of the MVTec AD dataset and their mean, comparing three binary map strategies: (a) THR ($\mu + 3\sigma$), (b) TTT4AS, and (c) TopoTTA. The table highlights the best result for each Precision, Recall, and F1 Score metric in **bold black** and the second-best in blue.

| Metric | Bottle | Cable | Capsule | Carpet | Grid | Hazelnut | Leather | MetalNut | Pill | Screw | Tile | T-brush | Transistor | Wood | Zipper | Mean |
|---|---|---|---|---|---|---|---|---|---|---|---|---|---|---|---|---|
| **(a) PaDiM - Binary Map - THR ($\mu + 3\sigma$) [71]** | | | | | | | | | | | | | | | | |
| **Precision** | 0.729 | 0.580 | 0.287 | **0.561** | 0.327 | **0.586** | **0.306** | 0.540 | **0.410** | 0.196 | 0.131 | **0.416** | 0.462 | **0.576** | 0.676 | 0.452 |
| **Recall** | 0.321 | 0.249 | 0.813 | 0.736 | 0.708 | 0.477 | 0.927 | 0.281 | 0.493 | 0.712 | 0.005 | 0.514 | 0.349 | 0.399 | 0.615 | 0.506 |
| **F1 Score** | 0.343 | 0.280 | 0.325 | 0.523 | 0.407 | 0.433 | **0.396** | 0.292 | 0.337 | 0.295 | 0.009 | 0.391 | 0.307 | 0.375 | 0.596 | 0.392 |
| **(b) PaDiM - Binary Map - TTT4AS [63]** | | | | | | | | | | | | | | | | |
| **Precision** | 0.585 | 0.412 | 0.176 | 0.429 | 0.199 | 0.349 | 0.208 | 0.519 | 0.269 | 0.088 | 0.137 | 0.258 | 0.472 | 0.355 | 0.499 | 0.330 |
| **Recall** | 0.438 | 0.500 | 0.707 | 0.769 | 0.726 | 0.637 | 0.916 | 0.491 | 0.568 | 0.735 | 0.123 | 0.595 | 0.425 | 0.416 | 0.648 | 0.579 |
| **F1 Score** | 0.429 | 0.395 | 0.214 | 0.459 | 0.290 | 0.376 | 0.293 | 0.386 | 0.262 | 0.153 | 0.103 | 0.283 | 0.291 | 0.319 | 0.512 | 0.317 |
| **(c) PaDiM - Binary Map - TopoTTA** | | | | | | | | | | | | | | | | |
| **Precision** | **0.750** | **0.648** | 0.355 | 0.523 | 0.463 | 0.358 | 0.246 | **0.574** | 0.307 | 0.266 | **0.685** | 0.268 | **0.492** | 0.439 | **0.678** | 0.470 |
| **Recall** | 0.689 | 0.670 | 0.828 | 0.942 | 0.805 | 0.885 | 0.987 | 0.636 | 0.783 | 0.905 | 0.742 | 0.920 | 0.547 | 0.756 | 0.724 | 0.787 |
| **F1 Score** | **0.718** | **0.658** | 0.496 | 0.672 | 0.587 | 0.509 | 0.393 | 0.603 | 0.441 | 0.411 | 0.712 | 0.415 | 0.518 | 0.555 | 0.700 | 0.559 |

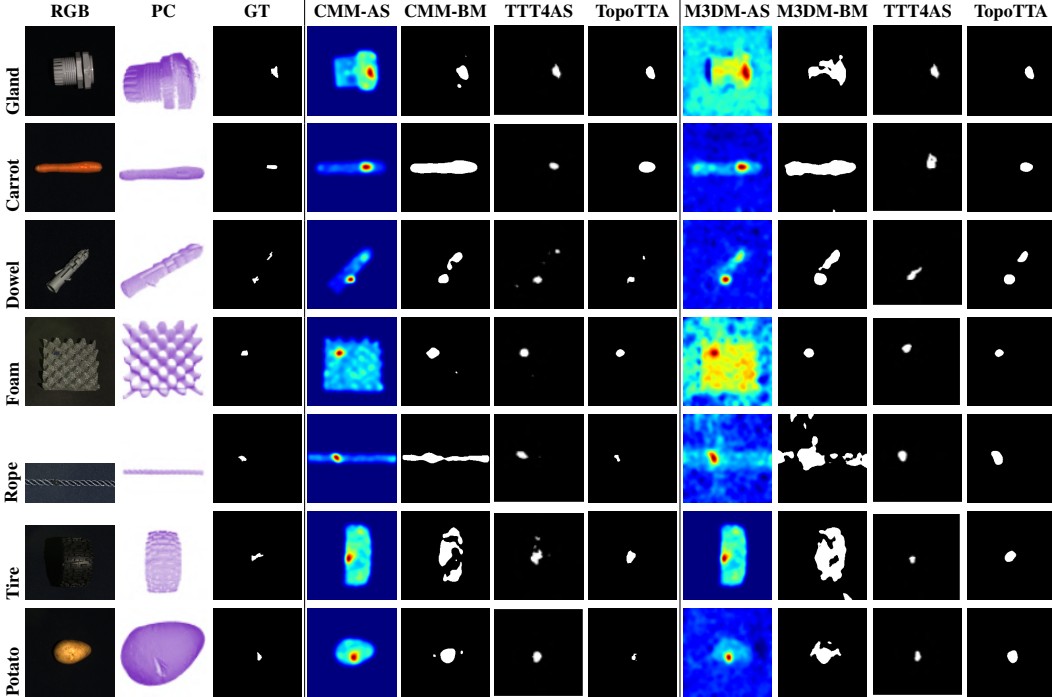

Figure 9: Qualitative comparison of AD&S methods for different objects using on 3D MvTec AD Dataset.

607 **A.6 Time Complexity Analysis**

608 The computational efficiency of *TopoTTA* is evaluated in terms of time complexity and GPU memory
609 utilisation. The architecture comprises two key components: the multi-level cubical complex filtration
610 module , which generates pseudo-labels using topological methods, and the PCES module , responsi-
611 ble for real-time anomaly detection. The filtration process introduces an overhead of approximately
612 0.1 seconds per image, primarily due to the use of CPU-based TDA libraries. While this represents a
613 current limitation, it also highlights the potential for significant speed improvements through future
614 GPU-accelerated implementations of cubical complex computations.

615 The PCES block demonstrates efficient inference performance, requiring only 0.14 seconds per
616 image, thereby contributing to the overall speed of the system. As a result, the total inference time
617 of the complete pipeline remains at approximately 0.23 seconds per image , which is competitive
618 with existing methods. For comparison, a standard 2D baseline model [71] reports an inference time
619 of 0.22 seconds per image, indicating that our method incurs minimal overhead despite the added
620 topological processing.

621 In the 3D domain, we benchmark our approach against state-of-the-art models. The M3DM [72]
622 model requires 2.86 seconds per image and consumes 6.52 GB of GPU memory, while the CMM
623 [73] model achieves faster inference at 0.12 seconds per image, using only 427 MB of GPU memory .
624 In contrast, our framework performs segmentation at 0.147 seconds per image with a modest GPU
625 memory usage of 1.52 GB, demonstrating a favorable balance between accuracy and efficiency. These
626 results highlight the resource effectiveness of our approach compared to existing 3D anomaly detection
627 models. Furthermore, they underscore the practical viability of incorporating TDA-based test-time
628 adaptation into real-world applications, especially as GPU support for topological computations
629 continues to evolve.

## A.7 Discussion, Limitations, and Future Directions

The *TopoTTA* framework introduces a novel perspective on TTA by incorporating PH into pseudo-label generation for AS. This topological guidance addresses a critical limitation of existing TTT-based AS methods, which often rely on handcrafted intensity thresholds or unsupervised peak suppression heuristics that do not generalise well across datasets or anomaly types. Our results demonstrate that by grounding the adaptation process in topological descriptors derived from cubical complex filtrations, *TopoTTA* achieves better segmentation performance, particularly in scenarios involving disconnected anomalies, irregular textures, or hollow defects.

A key insight emerging from our evaluations on MVTec 2D/3D and BraTS datasets is that structural consistency significantly improves model generalisability. This is evident in performance gains across multiple classes and modalities. Notably, the use of both sublevel and superlevel filtrations ensures robustness to both low- and high-intensity anomaly regions, which often correspond to different semantic manifestations (e.g., subtle scratches vs. severe cracks). Furthermore, the model-agnostic design of *TopoTTA* allows it to serve as a drop-in post-processor for a range of existing AD&S methods, enhancing segmentation without retraining or accessing source domain data.

Despite these promising results, the current formulation of *TopoTTA* comes with certain limitations. First, the quality of the refined segmentation mask is still constrained by the fidelity of the input anomaly score map $\Psi$. In cases where upstream AD&S models produce noisy or low-contrast heatmaps, such as in highly textured backgrounds or extremely subtle defects, the topological filtration process may fail to extract better persistent features. This highlights a broader dependency on the capacity of pre-trained feature extractors (e.g., DINO, PointMAE) and anomaly scorers (e.g., PatchCore, CMM), which limits performance in domains with poor feature transferability.

While *TopoTTA* generalises well across 2D and 3D modalities, it has not yet been extended to spatiotemporal data, such as anomaly segmentation in video or medical time-series imaging. These applications pose unique challenges, including temporal coherence, dynamic appearance variation, and online adaptation constraints, all of which require architectural extensions beyond static pseudo-label generation.

Moving forward, we envision several promising avenues to address these limitations and further develop the *TopoTTA* framework. First, we propose exploring differentiable approximations of persistent homology that would allow end-to-end training with topological loss functions. Recent work in this direction (e.g., differentiable persistence landscapes or vectorisations) could be integrated into *TopoTTA's* classifier training to dynamically align feature spaces with persistent structural signals.

Second, to mitigate the reliance on noisy anomaly scores, we plan to jointly optimise the anomaly map generation and topological filtering using self-supervised pretext tasks that are sensitive to geometric consistency. For example, learning topological contrastive embeddings could help denoise sparse score regions while still respecting the underlying anomaly structures. This may also reduce the method's dependence on the initial selection of pseudo-labels.

Third, we aim to extend *TopoTTA* for real-time, spatiotemporal inference tasks. This includes anomaly detection in video sequences, robotic inspection in industrial settings, and dynamic tumour segmentation in 4D MRI volumes. One potential direction is to evolve persistence diagrams across frames to enforce temporal consistency, tracking the lifespan and evolution of anomaly components over time.

Lastly, uncertainty-aware filtration strategies can be developed to explicitly quantify the confidence of topological features based on their persistence, variance across augmentations, or alignment with classifier uncertainty. This would allow *TopoTTA* to selectively adapt or abstain in ambiguous regions, a desirable property for safety-critical applications such as autonomous driving or medical diagnostics.

*TopoTTA* offers a principled and effective strategy for topology-aware test-time adaptation in anomaly segmentation. By leveraging multi-scale cubical filtrations and persistent homology, our method provides structurally stable pseudo-labels that guide lightweight classifier training without modifying the base network. Despite current limitations in differentiability and score quality dependence, our analysis and results suggest that *TopoTTA* lays a solid foundation for future research at the intersection of topological data analysis, unsupervised segmentation, and adaptive learning.

