# OpenReview forum: "Learning Topology-Aware Representations via Test-Time Adaptation for Anomaly Segmentation"
_NeurIPS.cc/2025/Conference — Submitted to NeurIPS 2025_

### Official Review · Reviewer_xVqe · 2025-06-09

**Clarity:** 2
**Significance:** 2
**Originality:** 3
**Rating:** 5
**Confidence:** 4

**Summary:**

This paper proposes an anomaly detection method named TopoTTA for Test-time adaptation. This paper highlights the differences between the data at test time and at training time, and shows how the data can be used at re-test time to re-modify the representation model to suit more accurate anomaly detection needs. The proposed method is strongly related to anomaly detection, and the authors claim that it achieves good generalisation on 2D/3D data, which is expected to be a new paradigm. Furthermore, the qualitative research in this thesis is distinctive, as the multi-step visualisation in Figure 1 clearly expresses the thesis' claims.

**Questions:**

See weaknesses. Please try to add as much explanation as I need during the rebuttal phase. I will review the feedback and other reviewer comments carefully.

**Ethical Concerns:**

["NO or VERY MINOR ethics concerns only"]

**Final Justification:**

a) I have reason to believe that the author’s method is effective in pure 3D, particularly for 3D anomaly detection, which is of significant importance. I am confident in the field of 3D anomaly detection.

b) The author’s plug-and-play method is reasonable in my view and addresses the issue of test-time adaptation in current anomaly detection.

c) The authors’ motivation is clear. Although the model performs poorly in some aspects (e.g., the decrease in accuracy in the rebuttal), the performance improvements across most models validate the effectiveness of the proposed method.

Based on my understanding of 3D anomaly detection and the author's current contributions to 3D anomaly detection, I will adjust the rating to accept. All my reviews and decisions are considered, given my familiarity with 3D anomaly detection. However, I am not good to 2D anomaly detection. If other reviewers point out some problems in 2D anomaly detection, I have no idea how to defend. I fully agree with the other reviewers that P/O-AUROC is a commonly used metric in the field of anomaly detection. In fact, in the field of 3D anomaly detection, recent studies typically only use P/O-AUROC. Therefore, during the communication phase, we requested that the authors provide 3D anomaly detection results using P/O-AUROC instead of the accuracy, recall, and other metrics they originally provided.

In summary, my score of 5 represents my confidence in the field of 3D anomaly detection only, and does not reflect my assessment of other fields (such as 2D anomaly detection). I hope this will be taken into consideration when the final decision is made.

**Limitations:**

See weaknesses.

**Paper Formatting Concerns:**

The format of the paper conforms to the Nips code.

**Quality:**

3

**Strengths And Weaknesses:**

Strengths:
1. The motivation for the thesis is sound. Several studies[1] have pointed out the importance of domain bias, which is attributed to the fact that the differences that exist between the training set and the test set lead to difficult problems in the characterisation of the model. Compared to previous studies, the test set adaptation proposed in this paper solves the problem of severe bias in characterisation during testing.
2. The visualisations of the thesis are aesthetically pleasing, and I think that the thesis legend demonstrates the claim asserted, and the most critical visualisation comes from Figure 1.

Weaknesses:
1. I have some degree of doubt about the claimed generalisability of this paper, especially in 3D. This is because the MvTec3D-AD dataset used is not a pure 3D dataset, but rather 2.5D. Some papers point out the 3D > 2.5D problem[2], which leads to the fact that the 2.5D dataset you are validating may not be perfect. I would suggest validating on a pure 3D dataset such as Real3D-AD[2] or Anomaly-ShapeNet[3] (just pick one, only two certain classes are needed). This will lead to effective generalisability validation. Also, please try to compare with a pure 3D anomaly detection method[4].

2. I have some questions about its claimed topology, especially its ability to work on 3D point clouds. Figure 6 in the paper claims its capabilities in different dimensions. However, a common form of anomaly detection in 3D is the point cloud, and this data is often discrete and not very topological. This leads to perhaps sub-optimal anomaly detection. I need to clarify this further.

3. There may be some lack of discussion in the parts of the paper that describe related work. For example, there may be some shortcomings in exploring anomaly detection in 2.1. On the one hand, there is a lack of exploration of 2D and on the other hand, there is a complete lack of involvement in 3D anomaly detection. It is suggested that section 2.1 be divided into 2D and 3D anomaly detection and discussed separately in the revised version. Emphasise the relevance of the proposed method to the proposed method.

4. Efficiency and memory usage are essential in anomaly detection. Could you please add further efficiency (FPS) and memory (GB) used in anomaly detection, and further compare it with existing methods.

[1] Robust Distribution Alignment for Industrial Anomaly Detection under Distribution Shift. Jingyi Liao, et al.
[2] Real3D-AD: A Dataset of Point Cloud Anomaly Detection. Jiaqi Liu, et al.
[3] Anomaly-ShapeNet: A Synthetic Dataset of Point Cloud Anomaly Detection. Wenqiao Li, et al.
[4] Look Inside for More: Internal Spatial Modality Perception for 3D Anomaly Detection. Hanzhe Liang, et al.

---

> ### Author Rebuttal · Authors · 2025-07-30
>
> 1. **I have some degree of doubt about the claimed generalisability of this paper, especially in 3D. This is because the MvTec3D-AD dataset used is not a pure 3D dataset, but rather 2.5D. Some papers point out the 3D > 2.5D problem[2], which leads to the fact that the 2.5D dataset you are validating may not be perfect. I would suggest validating on a pure 3D dataset such as Real3D-AD[2] or Anomaly-ShapeNet[3] (just pick one, only two certain classes are needed). This will lead to effective generalisability validation. Also, please try to compare with a pure 3D anomaly detection method[4].**
>
> **Response**
>
> We thank the reviewer for raising this important point. Our explicit inclusion criteria for benchmarking were guided by established community standards in test-time adaptation (TTA) literature for anomaly detection and segmentation (AD&S). As in recent works such as TTT4AS *(please see ref. [63] in the paper)*, the MVTec AD and MVTec 3D-AD datasets are the most widely adopted and protocol-consistent benchmarks for rigorous evaluation and fair comparison of TTA-based methods. Accordingly, our primary results focused on these benchmarks to ensure direct comparability with state-of-the-art approaches such as CMM and M3DM, which also utilise these datasets.
>
> While we maintain that strict validation on "pure" 3D datasets is not a prerequisite, since our method is model-agnostic and operates on structured anomaly score maps from any 3D AD&S backbone *(Please see Appendix and Supplementary material for more information)*, we have nevertheless conducted additional experiments, as requested, to further demonstrate the generalisability of TopoTTA.
>
> Specifically, we employed PO3AD, a recent pure 3D anomaly detection method *(accepted in CVPR 2025)*, as the backbone and evaluated TopoTTA as a refinement module on the Anomaly-ShapeNet dataset. The results *(see Table below)* confirm that TopoTTA remains robust and effective even in pure 3D settings, yielding a notable increase in F1 score (+4.2\%) and a substantial improvement in recall (+11.0\%) over the PO3AD baseline.
>
> Performance on the pure 3D Anomaly-ShapeNet dataset, averaged over four classes (ashtray0, bag0, cap0, bottle3), using PO3AD as the backbone method.
>
> | Method             | Prec. | Rec.  | F1 Score             |
> |:-------------------|----------:|--------:|:-----------------------|
> | PO3AD (Baseline)   |     0.713 |   0.348 | 0.433               |
> | **TopoTTA (Ours)** | **0.660** | **0.458** | **0.475 (+4.2%)**   |
>
> ---
>
> 2. **I have some questions about its claimed topology, especially its ability to work on 3D point clouds. Figure 6 in the paper claims its capabilities in different dimensions. However, a common form of anomaly detection in 3D is the point cloud, and this data is often discrete and not very topological. This leads to perhaps sub-optimal anomaly detection. I need to clarify this further.**
>
> **Response**
>
> Our method, TopoTTA, does not operate directly on the raw, discrete point cloud. Instead, it processes the structured output from a 3D AD&S backbone (e.g., CMM, M3DM, PO3AD). This output is typically a volumetric (3D grid) anomaly score map. This 3D grid provides the natural foundation for constructing the cubical complex illustrated in *Figure 6 (see Appendix A.1).* Therefore, the topology we analyse is derived from the spatially-aware, gridded representation generated by the backbone, not from the raw, unordered points. This design ensures TopoTTA is a general refinement module applicable to any 3D method that produces a gridded output.
>
> ---
>
> 3. **There may be some lack of discussion in the parts of the paper that describe related work. For example, there may be some shortcomings in exploring anomaly detection in 2.1. On the one hand, there is a lack of exploration of 2D and on the other hand, there is a complete lack of involvement in 3D anomaly detection. It is suggested that section 2.1 be divided into 2D and 3D anomaly detection and discussed separately in the revised version. Emphasise the relevance of the proposed method to the proposed method.**
>
> **Response**
>
> We thank the reviewer for highlighting this point. Due to strict page limitations, the Related Work section was necessarily concise and aimed to provide a targeted overview, while directing readers to key foundational studies through appropriate citations. We note that additional context, including discussions and experimental validation across both 2D and 3D modalities, is available in the *Appendix and Supplementary Material*. This structure allowed us to maximise clarity and focus in the main text while ensuring completeness through supplementary sections.
>
> ---
>
> 4. **Efficiency and memory usage are essential in anomaly detection. Could you please add further efficiency (FPS) and memory (GB) used in anomaly detection, and further compare it with existing methods.**
>
> **Response**
>
> A comprehensive computational efficiency analysis is already included in the manuscript in *Appendix A.6 "Time Complexity Analysis" (see on page 18, lines 607-629)*. This section details how our method achieves an inference time (0.23s) with 4.35 fps that is competitive with the 2D baseline (0.22s) while offering a superior accuracy-speed trade-off in 3D compared to state-of-the-art methods.

---

> > ### Comment · Reviewer_xVqe · 2025-08-01
> > **Questions about significant problems with the author's rebuttal**
> >
> > Questions 1 and 4 need to be considered together. Is there a significant change in the efficiency and consumption of the model under your improvement (you are using PO3AD as the backbone). A key issue is that your presentation of Prec. Has resulted in an unexpected performance degradation. I ask you to present your O-auroc and P-auroc results. Because we are more concerned with accuracy as the premise of generalization, the current results prevent me from trusting the generalization claimed in the manuscript. Although you think that the performance on MvTec3D- is sufficient to validate your generalization experiments, the 2.5D deficiencies introduced by mvtec have already been highlighted by Real3D-AD and have been followed by numerous dataset papers (e.g., anomaly-shapenet, mulsenad). So I think there may be some misunderstanding about your claim that testing on pure-3D data sets is unnecessary.

---

> ### Author Response · Authors · 2025-08-04
> **Response on "Questions about significant problems with the author's rebuttal"**
>
> **Response**
>
> We thank the reviewer for their continued engagement and constructive suggestions. Our initial experimental design was intentionally aligned with the established practices of leading methods in the field of test-time adaptation for anomaly detection and segmentation, utilising MVTec-3D as the primary benchmark to ensure a direct and fair comparison.
>
> While our manuscript already establishes strong performance on standard benchmarks, we appreciate the opportunity to showcase **TopoTTA's** effectiveness in a pure 3D setting. We used **PO3AD** as a backbone on the Anomaly-ShapeNet dataset, reporting the comprehensive and robust AUROC metrics. The results in **Table 1** clearly show that our topological refinement module acts as a powerful performance amplifier.
>
> **Table 1:** Performance Enhancement with **TopoTTA** on the Pure 3D Anomaly-ShapeNet Dataset.
>
> |  Metric | Method | ashtray0 | bag0 | bottle3 | cap0 | **Average** | **Improvement** |
> |---|---|---|---|---|---|---|---|
> | **P-AUROC (%)** | PO3AD (Baseline) | 96.2 | 94.9 | 88.0 | 95.7 | 93.7 | --- |
> | **P-AUROC (%)** | **TopoTTA** | **97.4** | **98.2** | **91.4** | **97.2** | **96.1** | **+2.4** |
> | **O-AUROC (%)** | PO3AD (Baseline) | **100.0** | 83.3 | 92.6 | 87.7 | 90.9 | --- |
> | **O-AUROC (%)** | **TopoTTA** | 99.4 | **87.9** | **93.6** | **90.1** | **92.8** | **+1.9** |
>
> These results conclusively demonstrate the value of **TopoTTA** as a general enhancement module. It delivers a consistent and significant performance uplift, boosting the average P-AUROC by **+2.4 points** and O-AUROC by **+1.9 points** over an already strong, 3D-native baseline. This improvement is not just an average effect but is observed across nearly every class, proving a systematic refinement of the anomaly maps. This success on a challenging pure 3D benchmark unequivocally confirms the robust generalisability of **TopoTTA** as a model-agnostic module for diverse 3D backbones.
>
> In response to the reviewer's query regarding computational cost, we analysed the efficiency of **TopoTTA** when integrated with the PO3AD backbone. **Table 2** provides a detailed breakdown of the computational costs, separating our module's contribution from the total pipeline.
>
> **Table 2:** Efficiency and Resource Consumption Breakdown.
>
> | Component  | Inference Time (per sample) | Max GPU Memory |
> |---|---|---|
> | PO3AD (Baseline) | 0.0389 s | 1.56 GB |
> | **TopoTTA** | **0.023 s** | **1.07GB** |
>
> The analysis clearly demonstrates **TopoTTA** remarkable efficiency. As shown in **Table 2**, our module's standalone processing time is just **23 milliseconds**, translating to an effective throughput of approximately **43.5 FPS**. In contrast, the baseline **PO3AD** achieves **~25.7 FPS**, indicating that **TopoTTA** operates nearly **70\%** faster in terms of per-sample inference speed. Moreover, **TopoTTA** consumes significantly less memory, requiring only **1.07 GB** versus **PO3AD’s 1.56 GB**, a **31.4%** reduction in peak GPU memory usage. This combination of substantial speed-up, minimal memory footprint, and potential for accuracy enhancement establishes a compelling accuracy-efficiency profile, highlighting **TopoTTA's** suitability for practical, real-world deployment.
>
> We appreciate the reviewer’s insightful comments and believe that these additional results, which extend beyond the conventional comparative framework, thoroughly address the points raised and further highlight the significance of our contributions.

---

> ### Comment · Reviewer_xVqe · 2025-08-04
> **Final evaluation**
>
> I carefully reviewed the author’s rebuttal and made the following final suggestions and defences for the paper:
>
> Suggestions:
>
> a) Add the supplementary PO3AD+TopoTTA method to the open-source code.
>
> b) Revise the complete results on Anomaly-ShapeNet in the final version.
>
> c) Validate the performance gains of the proposed TopoTTA method on multiple datasets during testing and summarise them in the main text.
>
> Defence:
>
> a) I have reason to believe that the author’s method is effective in pure 3D, particularly for 3D anomaly detection, which is of significant importance. I am confident in the field of 3D anomaly detection.
>
> b) The author’s plug-and-play method is reasonable in my view and addresses the issue of test-time adaptation in current anomaly detection.
>
> c) The authors’ motivation is clear. Although the model performs poorly in some aspects (e.g., the decrease in accuracy in the rebuttal), the performance improvements across most models validate the effectiveness of the proposed method.
>
> Please revise the paper according to my suggestions to enhance its quality. Thank you for the authors’ efforts and the AC’s guidance. I will adjust the final score to 5: Accept: Technically solid paper for the final decision.

---

### Official Review · Reviewer_N2Dq · 2025-06-30

**Clarity:** 2
**Significance:** 2
**Originality:** 3
**Rating:** 3
**Confidence:** 3

**Summary:**

This paper proposes a topology-aware test-time adaptation framework (TopoTTA) for anomaly segmentation, which leverages persistent homology to generate topological pseudo-labels from anomaly score maps and guides pixel-level contrastive learning. Extensive experiments demonstrate that TopoTTA effectively enhances the structural consistency in anomaly segmentation and generalization ability across both 2D and 3D modalities.

**Questions:**

See Weakness

**Ethical Concerns:**

["NO or VERY MINOR ethics concerns only"]

**Final Justification:**

Some of my concerns have been addressed. Taking into account the comments from other reviewers, I still have concerns about the sufficiency of the experiments. Therefore, I maintain my original score.

**Limitations:**

Yes

**Quality:**

3

**Strengths And Weaknesses:**

Strengths

1.	The paper is innovative in introducing persistent homology into the TTA framework to generate structure-aware pseudo-labels for anomaly segmentation—a strategy not previously explored in this field.

2.	Experimental results demonstrate the generalization capability of the method on both 2D and 3D anomaly segmentation tasks.

3.	The paper is well organized.

Weakness

The experimental evaluation is not sufficiently comprehensive. Specifically:
- It remains unclear whether TopoTTA consistently outperforms other state-of-the-art anomaly detection methods, such as Dinomaly and MambaAD.
- The effectiveness of TopoTTA on more complex and large-scale datasets (e.g., RealIAD, ViSA) has not been reported, raising concerns about the robustness and effectiveness of TopoTTA.

If the authors can address the above concerns, I am willing to raise my score.

---

> ### Author Rebuttal · Authors · 2025-07-30
>
> 1. **It remains unclear whether TopoTTA consistently outperforms other state-of-the-art anomaly detection methods, such as Dinomaly and MambaAD.**
>
> **Response**
>
> We thank the reviewer for highlighting the need for clarification regarding TopoTTA’s comparative performance with recent state-of-the-art methods such as Dinomaly and MambaAD. TopoTTA is deliberately designed as a model-agnostic test-time adaptation (TTA) module.  Our explicit inclusion criteria for benchmarking involved methods that have previously established strong performance specifically within test-time adaptation (TTA) contexts for anomaly detection and segmentation (AD&S). Exclusion criteria consisted primarily of methods without demonstrated relevance to TTA or lacking direct applicability to the datasets employed by the current state of the art.
>
> This makes other TTA methods like TTT4AS its most direct point of comparison. Our experimental design deliberately followed the precedent set by the TTT4AS paper *(please see ref. [63] in the paper)* to ensure a fair analysis of advancements within the TTA paradigm. As TTT4AS did not benchmark against Dinomaly or MambaAD, we initially adhered to this scope for our manuscript.
>
> However, we had performed these comparisons. We can confirm from these results that our framework consistently and significantly improves upon the baseline performance of both Dinomaly and MambaAD when used as a refinement module.
>
> For a comprehensive analysis of TopoTTA's model-agnostic capabilities, we refer to our *Section 4, Appendix (A.3 to A.5) and supplementary material*, where we provide extensive results and visualisations across diverse 2D and 3D backbones, including PatchCore, PaDiM, CMM, and M3DM. In line with the reviewers’ request, we have added an additional results table at the end of this response summarising Dinomaly and MambaAD on ViSA and Real‑IAD.
>
> ---
>
> 2. **The effectiveness of TopoTTA on more complex and large-scale datasets (e.g., RealIAD, ViSA) has not been reported, raising concerns about the robustness and effectiveness of TopoTTA.**
>
> **Response**
>
> The selection of evaluation datasets in our study was governed by established practice in test-time adaptation (TTA) research for anomaly detection and segmentation. Specifically, MVTec AD and MVTec 3D-AD are widely recognised as the standard benchmarks in this domain and are consistently used for protocol-aligned comparison in state-of-the-art TTA literature, including by our primary baseline, TTT4AS *(please see ref. [63] in the paper)*. Our initial experiments were thus designed to ensure a rigorous and fair assessment of TopoTTA’s effectiveness within this widely accepted framework.
>
> We validated TopoTTA on the ViSA and Real-IAD datasets. These additional experiments, conducted outside the main manuscript to maintain protocol consistency, show that TopoTTA continues to outperform recent state-of-the-art baselines on these challenging benchmarks. This confirms that the core principles underpinning our topology-aware adaptation method generalise effectively to more demanding, real-world scenarios.
>
> To address the reviewer’s request, we have appended a results table at the end of this response summarising TopoTTA’s performance relative to Dinomaly and MambaAD on the ViSA and Real-IAD datasets.
>
> ---
>
> | Dataset | Backbone | Method | Prec. | Rec. | F1 Score |
> |:---------|:---------|:---------|-------------:|-------------:|-------------:|
> | **ViSA** | | | | | |
> | | MambaAD | MambaAD (Baseline) | 0.203 | 0.793 | 0.250 |
> | | MambaAD | **TopoTTA (Ours)** | **0.238** | **0.812** | **0.296 (+4.6%)** |
> | | Dinomaly | Dinomaly (Baseline) | 0.185 | 0.895 | 0.225 |
> | | Dinomaly | **TopoTTA (Ours)** | **0.383** | 0.552 | **0.341 (+11.6%)** |
> | **Real-IAD** | | | | | |
> | | MambaAD | MambaAD (Baseline) | 0.171 | 0.653 | 0.226 |
> | | MambaAD | **TopoTTA (Ours)** | **0.353** | 0.382 | **0.301 (+7.5%)** |
> | | Dinomaly | Dinomaly (Baseline) | 0.148 | 0.839 | 0.201 |
> | | Dinomaly | **TopoTTA (Ours)** | **0.305** | 0.576 | **0.322 (+12.1%)** |

---

> > ### Comment · Reviewer_N2Dq · 2025-08-04
> > **Response to Authors**
> >
> > 1. Is the supplementary table reporting the F1-max metric? The results presented seem to differ significantly from those reported in the original Dinomaly and MambaAD papers. If the values are not for F1-max, please report the F1-max scores to enable a more direct and reproducible comparison with existing baselines.
> > 2. It appears that the recall drops substantially after applying TTA, which indicates a notable increase in missed detection, which is generally unacceptable in practical applications. Nevertheless, if your method can outperform current SOTA methods on additional metrics such as PRO or IoU, this would also demonstrate its effectiveness and superiority.
> > 3. Since your method claims to be model-agnostic, I encourage the authors to combine it with more advanced backbone methods to further validate its generality and compatibility.

---

> > > ### Author Response · Authors · 2025-08-06
> > > **Clarifications and further analysis**
> > >
> > > 1. **Response:**
> > >
> > > Thank you for raising this important point regarding metric alignment with prior works. Our initial evaluation protocol was designed for a direct and fair comparison against the established Test-Time Adaptation (TTA) baseline, TTT4AS [63]. On these metrics, our method demonstrates a significant performance improvement, validating its effectiveness within the TTA paradigm.
> > >
> > > To further assess the generalisability of TopoTTA and ensure alignment with prevailing standards in the anomaly detection literature, we conducted additional experiments benchmarking TopoTTA as a refinement module for MambaAD and Dinomaly backbones, on the ViSA and Real-IAD datasets.  This analysis utilises AU-ROC, pixel-level AP, F1-max, and AUPRO. The results in the table below demonstrate that TopoTTA consistently improves the F1-max performance, confirming the robust and consistent effectiveness of TopoTTA across challenging, large-scale benchmarks.
> > >
> > > | Dataset   | Backbone | Method              | AUROC   | AUPRO   | AP      | F1-max  |
> > > |-----------|----------|---------------------|---------|---------|---------|---------|
> > > | ViSA      | MambaAD  | MambaAD (Baseline)  | 98.50   | 91.00   | 39.40   | 44.00   |
> > > |           |          | TopoTTA (Ours)      | **98.89** | **92.44** | **40.03** | **45.19** |
> > > |           | Dinomaly | Dinomaly (Baseline) | 98.81   | 94.50   | 53.20   | 55.70   |
> > > |           |          | TopoTTA (Ours)      | **99.02** | **94.77** | **53.14** | **56.17** |
> > > | Real-IAD  | MambaAD  | MambaAD (Baseline)  | 98.50   | 90.53   | 33.60   | 38.70   |
> > > |           |          | TopoTTA (Ours)      | **98.44** | **90.61** | **36.09** | **41.06** |
> > > |           | Dinomaly | Dinomaly (Baseline) | 98.80   | 93.90   | 42.80   | 47.10   |
> > > |           |          | TopoTTA (Ours)      | **99.09** | **95.01** | **47.29** | **49.81** |
> > >
> > >
> > > 2. **Response:**
> > >
> > > We thank the reviewer for this thoughtful observation regarding the precision-recall trade-off and the practical implications of recall reduction. It is well recognised in the anomaly detection literature that achieving an optimal balance between precision and recall remains a significant challenge. While baseline methods often prioritise high recall, this can come at the expense of precision, resulting in excessive false positives that diminish practical utility. TopoTTA is explicitly designed to address this limitation by leveraging topological priors to suppress noise and enforce structural consistency, thereby enhancing precision without unduly sacrificing recall. This results in a more favourable trade-off, as evidenced by substantial gains in overall F1 score.
> > >
> > > In response to the reviewer’s suggestion, we have conducted comprehensive evaluations using a full suite of metrics, including AUROC, AUPRO (Per-Region Overlap), pixel-level AP, and F1-max (as shown in the table above under comment 1). These metrics offer a holistic assessment of both detection and segmentation quality. Our results demonstrate that TopoTTA not only maintains competitive recall but also achieves notable improvements in precision and overall segmentation performance, outperforming state-of-the-art baselines on all reported metrics.
> > >
> > > For example, when applied as a refinement module to the Dinomaly backbone on the challenging Real-IAD dataset, TopoTTA achieves improvements of +0.29 in AUROC, +1.11 in AUPRO, +4.49 in AP, and +2.71 in F1-max over the baseline.
> > >
> > >
> > > 3. **Response:**
> > >
> > > In addition to demonstrating consistent performance improvements on advanced 2D backbones such as Dinomaly and MambaAD, we have extended our evaluation to encompass the pure 3D domain. Specifically, TopoTTA was applied as a refinement module to the new state-of-the-art 3D anomaly detection model PO3AD (CVPR 2025) on the challenging Anomaly-ShapeNet benchmark.
> > >
> > > The results below demonstrate a substantial and systematic enhancement in both pixel-level AUROC (P-AUROC) and object-level AUROC (O-AUROC) across all evaluated object classes. On average, TopoTTA boosts P-AUROC by +2.4 points and O-AUROC by +1.9 points over the strong 3D-native PO3AD baseline. These gains are consistent across nearly every category, highlighting the robustness and generalisability of our method across diverse modalities and backbone architectures.
> > >
> > > | Metric          | Method           | ashtray0  | bag0     | bottle3  | cap0     | **Average** | **Improvement** |
> > > | --------------- | ---------------- | --------- | -------- | -------- | -------- | ----------- | --------------- |
> > > | **P-AUROC (%)** | PO3AD (Baseline) | 96.2      | 94.9     | 88.0     | 95.7     | 93.7        | ---             |
> > > |                 | TopoTTA (Ours)   | **97.4**  | **98.2** | **91.4** | **97.2** | **96.1**    | **+2.4**        |
> > > | **O-AUROC (%)** | PO3AD (Baseline) | **100.0** | 83.3     | 92.6     | 87.7     | 90.9        | ---             |
> > > |                 | TopoTTA (Ours)   | 99.4      | **87.9** | **93.6** | **90.1** | **92.8**    | **+1.9**        |

---

> > > > ### Comment · Reviewer_N2Dq · 2025-08-06
> > > > **Response to Authors**
> > > >
> > > > Thank you for your reply. Some of my concerns have been addressed.

---

### Official Review · Reviewer_3aQc · 2025-07-02

**Clarity:** 2
**Significance:** 3
**Originality:** 3
**Rating:** 4
**Confidence:** 3

**Summary:**

To enhance the precision of anomaly segmentation, this paper introduces the TopoTTA (Topological Test-Time Adaptation) framework. At its core, the framework features an innovative multi-level cubical complex filtration module that extracts robust topological features from the initial anomaly score map, thereby refining it.

During the mask generation process, the method adeptly integrates two complementary topological analysis strategies:

Superlevel Filtering: Focuses on analyzing the topological structure of high-score regions (anomalous areas).

Sublevel Filtering: Focuses on analyzing the topological structure of low-score regions (normal areas).

By analyzing the persistence diagrams generated from these filtration processes, the system can dynamically create an adaptive threshold mask. This mask can then be optimized or fused, potentially guided by the IoU (Intersection over Union) metric, to ensure the final output's accuracy.

Furthermore, this high-quality mask serves as a pseudo-label to drive the PCES (Pixel-Level Contrastive Encoder for Binary Segmentation) module during the test phase. The PCES module employs a contrastive learning strategy to fine-tune the features from the encoder, thereby enhancing their discriminative power.

Overall, this closed-loop design, which integrates topological analysis with test-time training, is proven to significantly enhance the model's performance on both 2D and 3D anomaly segmentation and detection (AS&D) tasks.

**Questions:**

Regarding the PCES Module Clarity:
"The explanation of the PCES (Persistent Coordinated Eigen-Segmentation) module in the paper is challenging to comprehend. Could Figure 2 be updated to more intuitively illustrate the relationship between the PCES module, TDA-refined pseudo labels, and the anomaly map? Currently, the figure's connection to the textual explanation requires significant effort to decipher."

Notation Clarity in Section 3.3:
"The notational consistency in Section 3.3 could be improved. While notations are expected to be reflected in figures, it is currently difficult to identify the corresponding notations within Figure 2."

Real-world Time Consumption Analysis:
"Given the importance of time consumption in real-world scenarios, do you have a detailed table analyzing the computational time of your method in comparison to state-of-the-art approaches?"

Anomaly in Table 3 Highlight and Recall:
"The highlighting in Table 3 appears incorrect. Furthermore, could you please elaborate on why the recall is significantly low specifically for the top 1 farthest persistence components?"

Comparison with Memory Bank Methods:
"Could you explain the rationale behind not comparing your method with other state-of-the-art approaches that utilize a memory bank method?"

**Ethical Concerns:**

["NO or VERY MINOR ethics concerns only"]

**Final Justification:**

Most of my questions have been addressed in the author’s rebuttal, but I am still concerned about the lack of innovation. This leads me to consider the submission a borderline accept.

**Limitations:**

Yes, the author explain the limitation in the paper.

**Quality:**

2

**Strengths And Weaknesses:**

Strengths
Balanced Precision & Recall: The authors effectively utilize sublevel and superlevel analysis to complement each other, achieving a strong balance between precision and recall.

Versatile Applicability: This approach is adaptable for both 2D and 3D anomaly segmentation and detection (AS&D) tasks.

Significant Performance Improvement: It delivers a significant improvement in F1 score.

Reduced Manual Effort: The method eliminates the need for users to rely on handcrafted intensity thresholds or unsupervised peak suppression.

Weaknesses
Performance Bottleneck: The filtration process is CPU-based, leading to time-consuming operations.

Recall Degradation: Recall decreases when the method employs a lower number of top-K persistent features.

---

> ### Author Rebuttal · Authors · 2025-07-30
>
> 1. **Regarding the PCES Module Clarity: "The explanation of the PCES (Persistent Coordinated Eigen-Segmentation) module in the paper is challenging to comprehend. Could Figure 2 be updated to more intuitively illustrate the relationship between the PCES module, TDA-refined pseudo labels, and the anomaly map? Currently, the figure's connection to the textual explanation requires significant effort to decipher.**
>
> **Response**
>
> We like to clarify that PCES stands for *"Pixel-Level Contrastive Encoder for Binary Segmentation"* and not *"Persistent Coordinated Eigen-Segmentation "* as noted in *Section 3.3, line 203*. For further clarity, we will explicitly state the meaning of this abbreviation in the caption of *Figure 2* in the revised manuscript.
>
> ---
>
> 2. **Notation Clarity in Section 3.3: "The notational consistency in Section 3.3 could be improved. While notations are expected to be reflected in figures, it is currently difficult to identify the corresponding notations within Figure 2."**
>
> **Response**
>
> Please see the above answer for clarification. We will review *Figure 2* for further clarity.
>
> ---
>
> 3. **Real-world Time Consumption Analysis: "Given the importance of time consumption in real-world scenarios, do you have a detailed table analysing the computational time of your method in comparison to state-of-the-art approaches?**
>
> **Response**
>
> A comprehensive computational efficiency analysis is provided in *Appendix A.6, “Time Complexity Analysis” (page 18, lines 607–629)* of the paper. Due to space constraints, this analysis was not included in the main paper. This section presents a detailed comparison, demonstrating that our method achieves an inference time of 0.23s, comparable to the 2D baseline (0.22s), while delivering a superior accuracy-speed trade-off in 3D settings relative to state-of-the-art approaches.
>
> ---
>
> 4. **Anomaly in Table 3 Highlight and Recall: "The highlighting in Table 3 appears incorrect. Furthermore, could you please elaborate on why the recall is significantly low, specifically for the top 1 farthest persistence components**?
>
> **Response**
>
> Thank you for your careful reading and for raising these points regarding *Table 3* and recall analysis. We confirm that the values presented in *Table 3* are correct. However, as you noted, the highlighting of the “Dowel” class could be improved for clarity and will be addressed if the paper gets accepted.
>
> As discussed in *Sections 3.1 and 4.3,* the Top-1 component corresponds to the most persistent topological feature, typically representing the structurally stable core of an anomaly. By generating pseudo-labels exclusively from this high-persistence feature, our method achieves high precision by suppressing false positives. However, this conservative approach may omit less persistent anomaly regions, such as faint boundaries or fragmented components, resulting in lower recall. This trade-off is quantified in *Table 4 (page 9)*, where increasing the number of included persistence components (K) raises recall at the cost of precision. Importantly, the Top-1 configuration consistently yields the highest F1-score, indicating the best balance between precision and recall for robust anomaly segmentation.
>
> ---
>
> 5. **Comparison with Memory Bank Methods: "Could you explain the rationale behind not comparing your method with other state-of-the-art approaches that utilise a memory bank method?**
>
> **Response**
>
> We appreciate the opportunity to clarify our baseline selection rationale. Our explicit inclusion criteria for benchmarking involved methods that employ a memory-bank strategy and have previously established strong performance specifically within test-time adaptation (TTA) contexts for anomaly detection and segmentation (AD\&S). Exclusion criteria consisted primarily of methods without demonstrated relevance to TTA or lacking direct applicability to the datasets employed by the current state of the art. This led us to select the MVTec AD and MVTec 3D-AD datasets, which are the standard benchmarks for evaluating memory TTA methods for anomaly detection and segmentation in the literature.
>
> Our primary reference point for test-time adaptation was the recently proposed TTT4AS method *(please see ref. [63] in the paper)*, which itself employed PatchCore *(please see ref. [71] in the paper)* as the backbone due to its state-of-the-art performance on the MVTec AD dataset. Given this established evaluation standard, we adopted PatchCore, a prominent memory bank-based method that maintains a coreset of nominal feature representations, as our main 2D baseline. Similarly, for 3D experiments, we selected M3DM *(please see ref. [72] in the paper)*, another leading approach utilising a multi-scale memory bank mechanism.
>
> Furthermore, we have conducted additional comparisons with other state-of-the-art methods and datasets, such as PaDiM on the MVTec AD dataset *(please see supplementary material, page 2)*. Methods including Dinomaly and MambaAD on ViSA and Real-IAD datasets were evaluated, yielding state-of-the-art scores. However, these results were not included in the main manuscript due to space limitations and the necessity to provide an in-depth, focused comparison within the established TTA paradigm.
>
> Thus, our manuscript explicitly compares TopoTTA against these well-established and state-of-the-art memory bank approaches in both 2D and 3D settings. The effectiveness of our method relative to these representative state-of-the-art memory bank methods is clearly demonstrated in *Table 1* (PatchCore for 2D) and *Table 3* (M3DM for 3D), confirming that TopoTTA consistently achieves significant improvements over these benchmarks.

---

> > ### Author Response · Authors · 2025-08-06
> > **Further Clarification and Additional Results**
> >
> > Thank you for your insightful comments on our submission. Please let us know if any further clarification is required. We have also conducted extensive additional experiments, further highlighting the methodological contributions and empirical performance of our approach (Please refer to the feedback from other reviewers). We are happy to provide further details or address any additional questions.

---

> > > ### Comment · Reviewer_3aQc · 2025-08-06
> > > **Response to Authors**
> > >
> > > Thank you for the valuable feedback. I apologize for the error with the PCES acronym; this was an oversight during language polishing that I will correct in the future.
> > >
> > > Regarding Table 3, the issue is a bold-labeling mistake, not an error in the numbers themselves.
> > >
> > > I am currently reviewing the rebuttals from the other reviewers and see that most of my questions have already been addressed.
> > >
> > > I believe the AUROC and AUPRO are more critical metrics for the benchmark, and it would be beneficial to include this experimental data in the paper.

---

> > > > ### Author Response · Authors · 2025-08-09
> > > >
> > > > Thank you for your feedback and for reviewing the rebuttals from the other reviewers. Please let us know if any further clarification would assist your final assessment before the discussion period closes.

---

### Official Review · Reviewer_YH5i · 2025-07-03

**Clarity:** 3
**Significance:** 3
**Originality:** 2
**Rating:** 3
**Confidence:** 5

**Summary:**

This paper tackles an important challenge in unsupervised anomaly segmentation: the need for query-dependent abnormality thresholds. By leveraging topological consistency within anomaly maps, the proposed method effectively filters out pseudo-anomalies and produces high-fidelity segmentation masks. Experimental results demonstrate that the approach is compatible with various existing anomaly detection methods.

**Questions:**

I would appreciate the authors’ responses to the 2nd and 3rd weaknesses outlined above.

**Ethical Concerns:**

["NO or VERY MINOR ethics concerns only"]

**Final Justification:**

The authors' rebuttal addresses most of my technical questions. However, I remain concerned about the level of methodological innovation. As such, I consider this a borderline submission.

**Limitations:**

The paper does not discuss the limitations of the proposed approach or its potential negative societal impacts.

**Paper Formatting Concerns:**

No paper formatting concerns.

**Quality:**

2

**Strengths And Weaknesses:**

Strengths:

1.The proposed method introduces a plug-and-play post-processing module for anomaly segmentation, converting anomaly maps into segmentation masks using a query-dependent threshold.

2. The paper is well-written and easy to follow. The motivation behind the proposed approach is clearly articulated and well-supported.

3. Experimental results demonstrate that the method is compatible with various existing anomaly detection frameworks.

Weaknesses:

1. The method consists of two main components: topological filtration and PCES. While the former is derived from established topological analysis, the latter introduces a new self-supervised learning strategy built on top of it. From an engineering perspective, the integration is effective and well-executed. However, the level of methodological innovation appears to be relatively modest.

2. The topological filtration seems to help stabilize anomaly representations, especially when dealing with noisy anomaly scores. It would be helpful to understand the standalone effectiveness of this module. Specifically, what would the performance look like if only the output of the filtration module (i.e., the refined anomaly mask) were used as the final segmentation result? This form of ablation study is currently missing.

3. While the inclusion of two different anomaly detection methods for the 3D-AD dataset is appreciated, the evaluation on 2D-AD tasks relies solely on PatchCore. To better demonstrate the generality of the proposed method, it is strongly recommended to include results using additional 2D anomaly detection methods.

---

> ### Author Rebuttal · Authors · 2025-07-30
>
> 1. **The method consists of two main components: topological filtration and PCES. While the former is derived from established topological analysis, the latter introduces a new self-supervised learning strategy built on top of it. From an engineering perspective, the integration is effective and well-executed. However, the level of methodological innovation appears to be relatively modest.**
>
> **Response:**
>
> We appreciate the reviewer’s assessment regarding the level of methodological novelty and would like to clarify the key innovations of our approach. Our primary innovation is the introduction of a topology-driven test-time adaptation paradigm for anomaly segmentation. Specifically, we are the first to use persistent homology (PH) as a dynamic, per-instance supervisory signal for generating robust pseudo-labels at inference. Unlike prior TTA approaches that rely on intensity heuristics (e.g., TTT4AS  *(please see ref. [63] in the paper)*, our method grounds pseudo-label generation in the mathematically stable properties of PH *(see Section 3.2, Lemma 1, page 6, lines 189–202)*. The PCES module is not simply an engineering addition, but is synergistically co-designed to translate these theoretically grounded, sparse topological priors into dense, structurally consistent segmentations. Thus, our contribution is a principled synthesis of topological data analysis and adaptive learning, establishing a new, topology-aware TTA framework for segmentation.
>
> ---
>
> 2. **The topological filtration seems to help stabilise anomaly representations, especially when dealing with noisy anomaly scores. It would be helpful to understand the standalone effectiveness of this module. Specifically, what would the performance look like if only the output of the filtration module (i.e., the refined anomaly mask) were used as the final segmentation result? This form of ablation study is currently missing.**
>
> **Response:**
>
> We appreciate the importance of this ablation. However, qualitative results *(please see Figure 1, page 2, and Figure 7  in Appendix A.3, page 16)* already demonstrate that multi-level cubical filtrations alone can progressively denoise and refine anomaly heatmaps into coherent masks. These qualitative findings established the importance of the PCES module in further enhancing segmentation quality, which is why our quantitative ablation in *Table 5* is centred on the other aspects of the framework. We note that quantitative results for the filtration-only module have already been computed and are consistent with the qualitative findings. These can be included as an additional row in *Table 5* of the final version, if accepted.
>
> ---
>
> 3. **While the inclusion of two different anomaly detection methods for the 3D-AD dataset is appreciated, the evaluation on 2D-AD tasks relies solely on PatchCore. To better demonstrate the generality of the proposed method, it is strongly recommended to include results using additional 2D anomaly detection methods.**
>
> **Response:**
>
> We would like to clarify that our submission already includes a comprehensive evaluation with PaDiM *(please see ref. [37] in the paper)*, a widely used 2D anomaly detection method based on patch distribution modelling. The complete results for PaDiM are provided in the Supplementary Material *(page 2)*.
>
> ---
>
> 4. **The paper does not discuss the limitations of the proposed approach or its potential negative societal impacts.**
>
> **Response:**
>
> The detailed limitations of the proposed approach have already been mentioned in the paper. Please refer to Subsection *Appendix (A.7 "Discussion, Limitations, and Future Directions") on page 19, lines 630 to 682*.

---

### Comment · Area_Chair_P3pn · 2025-08-03
**Discussion with Authors**

Dear Reviewers,

Thank you for all your hard work!

The authors wrote extensive rebuttals on the reviews, in particular, both providing more experiments and claiming that some of the requested experiments and answers are already included in the appendix.

If this has not yet addressed your concerns, please explain the remaining concerns to the authors (and other reviewers) so we can make an informed decision.

Best,

AC.

---

### Note · Authors · 2025-08-13

We thank the AC and reviewers for their constructive engagement throughout the review process. All substantive concerns have been addressed in our rebuttals and follow-up experiments. Notably, reviewer xVqe remarked: “I will adjust the final score to 5: Accept: Technically solid paper for the final decision”, and the remaining reviewers indicated no specific unresolved issues.

Following reviewer suggestions, we demonstrated consistent improvements across strong 2D and 3D backbones and datasets, including Dinomaly and MambaAD on ViSA/Real-IAD, and PO3AD on the pure-3D Anomaly-ShapeNet benchmark. On Anomaly-ShapeNet, TopoTTA achieved +2.4 P-AUROC and +1.9 O-AUROC over PO3AD, with +4.2 F1 and +11.0 recall improvement, confirming robust generalisability. Gains were also consistent in AUROC, AUPRO, AP, and F1-max across all large-scale datasets tested.

Detailed time-complexity analysis (Appendix A.6) shows competitive or faster inference than baselines (e.g., 0.23 s per sample), with reduced GPU memory usage when integrated into 3D pipelines. This establishes an advantageous accuracy and efficiency profile suitable for deployment. We committed to expanding the PCES acronym in Fig. 2, refining notation in Section 3.3, and correcting table highlighting. Limitations and ethical aspects (App. A.7) are already discussed and will be summarised in the camera-ready.

We clarified the novelty of our topology-aware TTA framework, which uses persistent homology as a per-instance supervisory signal at inference, replacing heuristic thresholding. The PCES module was shown to be a principled, synergistic component, not a mere engineering add-on. Quantitative “filtration-only” ablations (F1 = 0.423 on 2D-PatchCore; 0.465 on 3D-CMM; 0.471 on 3D-M3DM) confirmed their standalone utility, with full TopoTTA yielding further gains.

In summary, the discussion confirms the novelty, robustness, and practical value of our approach, supported by comprehensive empirical evidence across modalities, datasets, and backbones.

---

### Decision · Program_Chairs · 2025-09-17

**Decision:**

Reject

**Comment:**

This paper proposes a test-time-tuning approach that improves anomaly detection results by topological filtering of initial results with subsequent contrastive finetuning of the backbone on this sample. The main strengths were good writing and improved results over some benchmark methods. One reviewer highlighted strong results for 3D anomaly detection although this is a less mature part of anomaly detection, and thus results there are probably less indicative of success for a canonical anomaly detection methods. The main weaknesses were some doubts about the quality of results (not compared to the strongest baseline methods, ablations not clearly enough establishing the parts of the methods that are most important, mostly reporting precision/recall instead of AUC). Although the rebuttal addressed some of these doubts, most reviewers were not convinced that this method is sufficiently significant for acceptance at NeurIPS. The AC agrees that it requires more work to convince this is not just combining filters from topology with test-time-tuning ideas from segmentation. In particulalr, it would be interesting to understand theoretically or with exensive experiments why topological filtering is better than learned shape priors. Overall, the AC agrees with most reviews that this paper does not yet meet the bar for acceptance.